

**Link between the Outgoing Longwave Radiation and the**
**altitude where the space-borne lidar beam is fully attenuated**
Thibault Vaillant de Guélis[1], Hélène Chepfer[1], Vincent Noel[2], Rodrigo Guzman[3], Philippe Dubuisson[4],
David M. Winker[5], Seiji Kato[5]
[1]LMD/IPSL, Université Pierre et Marie Curie, Paris, France
[2]Laboratoire d'Aérologie, CNRS, Toulouse, France
[3]LMD/IPSL, CNRS, École polytechnique, Palaiseau, France
[4]Laboratoire d'Optique Atmosphérique, Université Lille, Lille, France
[5]NASA Langley Research Center, Hampton, Virginia, USA
*Correspondence to*: Thibault Vaillant de Guélis (thibault.vaillant-de-guelis@lmd.polytechnique.fr)
**Abstract.** According to climate models' simulations, cloud altitude change is the dominant contributor of the positive
ensemble mean longwave cloud feedback. Nevertheless, the cloud altitude longwave feedback mechanism and its amplitude
struggle yet to be verified in observations. An accurate, stable in time, and potentially long-term observation of a cloud
property summarizing the cloud vertical distribution and driving the longwave cloud radiative effect is needed to hope to
achieve a better understanding of the cloud altitude longwave feedback mechanism. This study proposes the direct lidar
measurement of the atmosphere opacity altitude is a good candidate to derive the needed observed cloud property. This
altitude is the level at which a space-borne lidar beam is fully attenuated when probing an optically opaque cloud. By
combining this altitude with the direct lidar measurement of the cloud top altitude, we derive the radiative temperature of
opaque clouds that linearly drives, as we show, the outgoing longwave radiation. This linear relationship provides a simple
formulation of the cloud radiative effect in the longwave domain for opaque clouds and so, helps to understand the cloud
altitude longwave feedback mechanism. We find that in presence of an opaque cloud, a cloud temperature change of 1 K
modifies its cloud radiative effect by 2 W·m⁻². We show that this linear relationship holds true at single atmospheric column
scale with radiative transfer simulations, at instantaneous radiometer footprint scale of the Clouds and the Earth's Radiant
Energy System (CERES), and at monthly mean 2°×2° gridded scale. Opaque clouds cover 35 % of the ice-free ocean and
contribute to 73 % of the global mean cloud radiative effect. Thin clouds cover 36 % and contribute to 27 %.



**1 Introduction**

Cloud feedback mechanisms remain the main source of uncertainty for current predictions of the climate sensitivity

(e.g. Dufresne and Bony, 2008; Vial et al., 2013; Webb et al., 2013; Caldwell et al., 2016). Clouds simulated by climate
models in the current climate, exhibit large biases compared to observations (e.g. Zhang et al., 2005; Haynes et al., 2007;
Chepfer et al., 2008; Williams and Webb, 2009; Marchand and Ackerman, 2010; Cesana and Chepfer, 2012; Kay et al.,
2012; Nam et al., 2012; Cesana and Chepfer, 2013; Klein et al., 2013) leading to low confidence in the cloud feedbacks
predicted by climate models.

In order to understand the feedback mechanisms, it is useful to identify the fundamental variables that drive the

climate radiative response, and then decompose the overall radiative response as the sum of individual radiative responses
due to changes in each of these variables. This classical feedback analysis has been largely applied to outputs from numerical
climate system simulations in order to estimate the effects of water vapor, temperature lapse rate, clouds and surface albedo
on the overall climate radiative response (e.g. Cess et al., 1990; Le Treut et al., 1994; Watterson et al., 1999; Colman, 2003;
Bony et al., 2006; Bates, 2007; Soden et al., 2008; Boucher et al., 2013; Sherwood et al., 2015; Rieger et al., 2016). Focusing
only on the cloud feedback mechanisms, such approach (Zelinka et al., 2012a) has been used to isolate the role of each
fundamental cloud variables that contribute to the radiative response: the cloud cover, the cloud optical depth or condensed
water (liquid and ice), and the cloud altitude (or cloud temperature). The shortwave (SW) cloud feedback is driven by
changes in the cloud cover and the cloud optical depth, whereas the longwave (LW) cloud feedback is driven by changes in
the cloud cover, the cloud optical depth and the cloud vertical distribution (e.g. Klein and Jakob, 1999; Zelinka et al., 2012a,
2012b, 2013).

Verifying cloud feedback mechanisms that have been predicted by climate models simulations using observations

requires two steps: 1) First, establish a direct and robust link between the observed fundamental cloud variables and the
cloud radiative effet (CRE) at the top of the atmosphere (TOA); so that any change in the fundamental cloud variables can be
unambiguously translated within a change in the CRE at the TOA, 2) Second, establish an observational record of these
cloud fundamental variables that is long enough, stable enough and accurate enough to detect cloud changes due to
greenhouse gases forcing (Wielicki et al., 2013). Such records do not exist yet. Despite this last limitation, Klein and Hall
(2015) suggested that some cloud feedback mechanisms, namely the "emergent constraints", could be tested with shorter
records in comparing the simulated and the observed current climate interannual variabilities.

The current paper focuses on the LW cloud feedback. Current climate models consistently predict that the cloud

altitude change is the dominant contributor to the LW cloud feedback (Zelinka et al., 2016) in agreement with previous
works (e.g. Schneider, 1972; Cess, 1975; Hansen et al., 1984; Wetherald and Manabe, 1988; Cess et al., 1996; Hartmann and
Larson, 2002). If the models agree on the sign and the physical mechanism of the LW cloud altitude feedback, they predict
different amplitudes. Coupled Model Intercomparison Project Phase 5 (CMIP5) climate model simulations suggest that the
cloud altitude would rise up by 0.7 to 1.7 km in the upper troposphere in all regions in a warmer climate (+4 K), which is a
significant change compared to the currently observed variability, and thus, could be a more robust observable signature of
climate change than the CRE (Chepfer et al., 2014). Nevertheless, the cloud altitude LW feedback mechanism and its
amplitude still struggle to be verified in observations. There is still no observational confirmation for the altitude LW cloud
feedback mechanism because 1) there is no simple direct and robust formulation linking the observed fundamental cloud
variables and the LW CRE at the TOA 2) there is no accurate and stable observations of the vertical distribution of clouds
over several decades.

Thus, a preliminary step to progress on the LW cloud feedback is to establish a direct and robust link between the

LW CRE at the TOA and fundamental cloud properties that can be accurately observed and which can also be simulated in
climate models. In the SW, Taylor et al. (2007) defined such a simplified radiative transfer model by robustly expressing the
SW CRE as a function of the cloud cover and the cloud optical depth. This linear relationship has been largely used for




decomposing the SW cloud feedbacks into contributions due to cloud cover change and optical depth change. Contrary to the
SW, the LW CRE does not only depend on the cloud cover and the cloud optical depth, but also on the cloud vertical
distribution. As stated in Taylor et al. (2007) and in the attempt made by Yokohata et al. (2005), establishing a simple
radiative transfer model that robustly expresses the LW CRE as a function of a limited number of properties (which can be
reliably observed and which can also be simulated in climate models), is more challenging in the LW than in the SW because
the LW involves three variables instead of two: the cloud cover, the cloud optical depth and the cloud vertical distribution.

Complete radiative transfer simulations allow to accurately compute the LW CRE for a well-defined atmosphere

(clear sky and clouds): detailed information on the atmospheric columns collected by active sensors have been used to
estimate TOA CRE and surface CRE (e.g. Zhang et al., 2004; L'Ecuyer et al., 2008; Kato et al., 2011; Rose et al., 2013). In
contrast, the definition of a simple and robust linear formulation between the LW CRE at the TOA and a limited number of
cloud variables, that would be useful for climate cloud feedback decomposition, cannot use the details of the entire cloud
vertical distribution: first, one needs to summarize the entire cloud vertical profile within a few specific cloud levels that
drives the LW CRE at the TOA, and second, this specific cloud levels need to be accurately observed at global scale from
satellites.

Most of the cloud climatologies derived from space observations rely on passive satellites, which do not retrieve the

actual cloud vertical distribution, and only retrieve the cloud top pressure and estimates of high-level, mid-level, and low-
level cloud covers. These last estimates have been coupled with ranges of cloud optical depth to define different cloud types
(Hartmann et al., 1992) associated to different values of CRE. These cloud types have been used to analyze the interannual
cloud record collected by the Moderate Resolution Imaging Spectroradiometer (MODIS) (Zelinka and Hartmann, 2011), as
well as the International Satellite Cloud Climatology Project (ISCCP) and the Pathfinder Atmospheres Extended (PATMOS-
x) (Marvel et al., 2015; Norris et al., 2016) in order to identify LW CRE changes associated to cloud properties changes.

But recently, Stephens et al. (submitted) used combined passive observations and active sensors observations (2B-

FLXHR-LIDAR product; Henderson et al., 2013) collected by the Cloud-Aerosol LIdar with Orthogonal Polarization
(CALIOP) from the Cloud-Aerosol Lidar and Infrared Pathfinder Satellite Observations (CALIPSO) and the Cloud Profiling
Radar (CPR) from CloudSat (Stephens et al., 2002) to re-build similar cloud types as in Hartmann et al. (1992). Stephens et
al (submitted) and Hartmann et al. (1992) found very different results because passive sensors cannot retrieve reliable cloud
altitude contrarily to active sensors (e.g. Sherwood et al., 2004; Holz et al., 2008; Michele et al., 2013; Stubenrauch et al.,
2013). Today, ten years of satellite-borne active sensors data provide a detailed and accurate view of the cloud vertical
distribution, which can be used to build for the first time, a simplified radiative transfer model that robustly expresses the
LW CRE as a function of the cloud cover, the optical depth (or emissivity) and the cloud altitude, and that can be tested
against observations. To do so, in the current paper, we summarize the entire cloud vertical profile observed by active
sensors with three specific cloud levels that drive the LW CRE at the TOA and that can be accurately observed by space-
borne lidar: the cloud top altitude, the cloud base altitude, and the altitude of opacity where the laser beam gets fully
attenuated when it passes through an Opaque cloud. This altitude of opacity together with the Opaque cloud cover, are both
observed by space-borne lidar, and are strongly correlated to the LW CRE (Guzman et al., 2017) because emissions of layers
located below the altitude of opacity have little influence on the outgoing LW radiation (OLR). Previous studies
(Ramanathan, 1977; Wang et al., 2002), suggested that the link between the Opaque cloud temperature and the OLR is
linear, which would be mathematically very convenient for the study of cloud feedbacks (derivatives), but these studies are
limited to radiative transfer simulations only. We propose to build on these studies by adding the space-borne lidar
information.

In Section 2 we present the data and tools used in this study. In Section 3 we define radiative temperatures of

Opaque clouds and Thin clouds derived from lidar cloud altitude observations and reanalysis, and present the observed
distributions over the mid-latitudes region and the ascending and subsiding regime areas in the tropics. In Section 4 we use



radiative transfer simulations to establish a simple expression of the OLR as a function of lidar cloud observations for
Opaque cloud single columns, and for Thin cloud (non-opaque) single columns by adding the Clouds and the Earth's Radiant
Energy System (CERES) clear sky observations. Then, we verify this relationship against observations at instantaneous
20 km scale, using high spatial resolution collocated satellite-borne broadband radiometer (CERES) and lidar data
(CALIPSO), and at monthly mean 2° latitude × 2° longitude gridded scale. In Section 5 we estimate the independent
contributions of optically Opaque clouds and optically Thin clouds to the CRE. We then focus on the Tropics and examine
Opaque and Thin cloud CREs partition in subsidence and deep convective regions. Section 6 discusses the limits of the
linear expression we propose, and concluding remarks are summarized in Section 7.




## 124   2 Data and Tools

### 125   2.1 Opaque and Thin clouds observations by space-borne lidar

Eight years (2008–2015) of CALIPSO observations are used in this study. The GCM-Oriented CALIPSO Cloud
Product (GOCCP)-OPAQ (GOCCP v3.0; Guzman et al., 2017) segregates each atmospheric single column sounded by the
CALIOP lidar as one of the 3 following single column types (Fig. 1):
• The *Clear sky single column* (brown, center) is entirely free of clouds. In other words, none of the 40 levels of
480 m vertical resolution composing the atmospheric single column is flagged as "Cloud" (Chepfer et al., 2010).
• The *Opaque cloud single column* (orange, right) contains a cloud into which the laser beam of the lidar ends fully
attenuated at an altitude termed $Z^|_{Opaque}$. $Z^|_{Opaque}$ (as well as any $X^|$ variable used later on in the paper) refers to a
*single column*, i.e. a 1D atmospheric column from surface to the TOA where each altitude layer is homogeneously
filled with molecules and/or clouds, as mentioned by the exponent symbol "|". Such single column is directly
identified by the presence of a level flagged as "z_opaque". Full attenuation of the lidar is reached for a visible
optical depth, integrated from the top of the atmosphere (TOA), of about 3 to 5 (Vaughan et al., 2009). This
corresponds to a cloud LW emissivity of 0.8 to 0.9, if we consider that cloud particles do not absorb visible
wavelengths and that diffusion can be neglected in the LW domain.
• The *Thin cloud single column* (brown and blue, left), contains a semi-transparent cloud. Such single column is
identified by the presence of at least one level flagged as "Cloud" without a level flagged as "z_opaque".


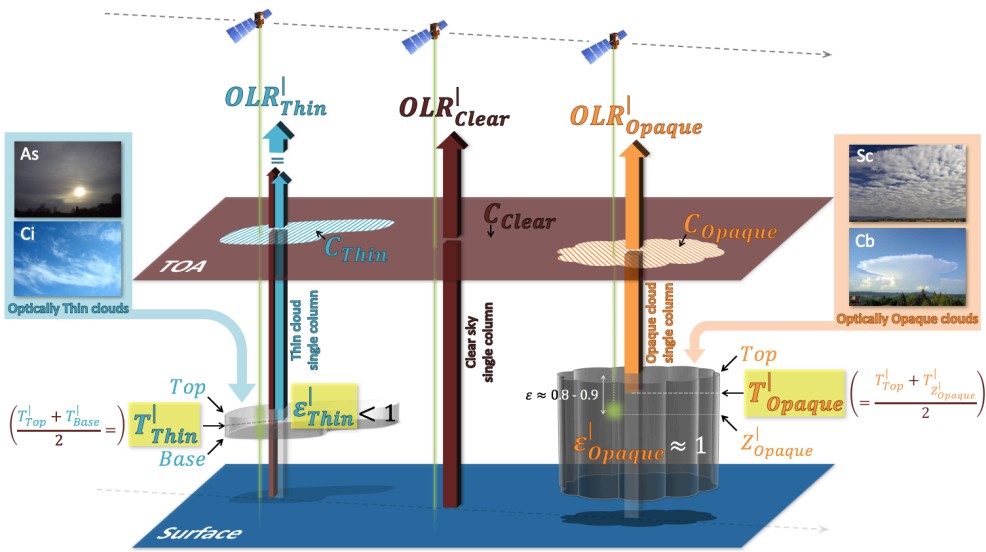

FIG. 1. Partitioning of the atmosphere into 3 single column types thanks to CALIOP lidar: (left) Thin cloud single column, when a
cloud is detected in the lidar signal and the laser beam achieve to wholly go through the cloud until the surface, (middle) Clear sky single
column, when no cloud is detected, and (right) Opaque cloud single column, when a cloud is detected and the laser beam ends fully
attenuated into the cloud at a level called $Z^|_{Opaque}$. $C$, $T$ and $\varepsilon$ respectively account for cover, temperature and emissivity. The variables
highlighted in yellow are the key cloud properties, extracted from the GOCCP-OPAQ product, that drive OLR over Thin cloud and
Opaque cloud single columns. The total gridded OLR will be computed from the 3 single column OLRs weighted by their respective
cover: $C_{Thin}$, $C_{Clear}$, $C_{Opaque}$.

Figure 2 shows the global covers of these 3 single column types, using 2°×2° grids. The global mean Opaque clouds
cover $C^{\boxplus}_{Opaque}$ is 35 %, Thin clouds cover $C^{\boxplus}_{Thin}$ is 36 % and the Clear sky cover $C^{\boxplus}_{Clear}$ is 29 %. $C^{\boxplus}_{Opaque}$, $C^{\boxplus}_{Thin}$ and $C^{\boxplus}_{Clear}$





(as well as any $X^{\boxplus}$ variable used later on in the paper) refer to 2°×2° grid box as mentioned by the exponent symbol "⊞".
Opaque clouds cover is very high at mid-latitudes and, in the tropics, high occurrences clearly reveal regions of deep
convection (warm pool, ITCZ) and stratocumulus regions at the east part of oceans. Thin clouds cover is very homogeneous
over all oceans, with some slight maxima in some regions, namely near the warm pool. These results are discussed in detail
in Guzman et al. (2017).

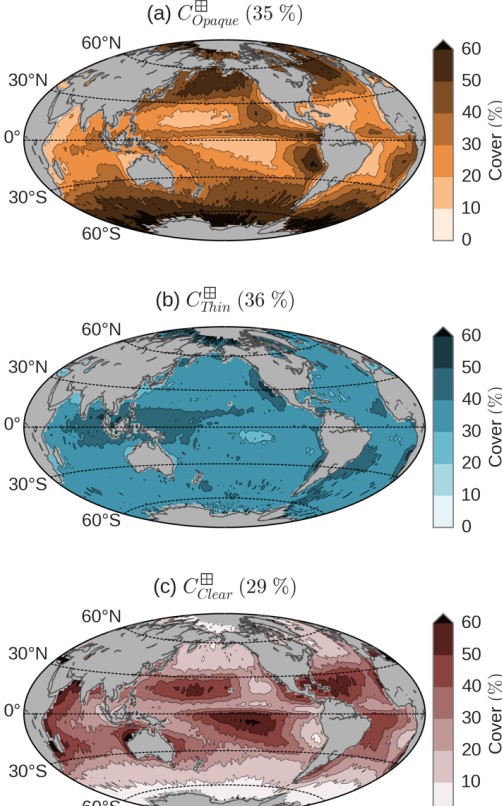

FIG. 2. Maps of (a) Opaque cloud cover (b) Thin cloud cover and (c) Clear sky cover. Only nighttime over ice-free oceans for the 2008–2015 period is considered. Global mean values are given in parentheses.



Our study builds on Guzman et al. (2017) by using temperatures instead of altitudes, and by estimating an
additional variable, the Thin cloud emissivity:
•   Temperatures $T^{|}_{Z^{|}_{Opaque}}$, $T^{|}_{Top}$ and $T^{|}_{Base}$ are respectively those at the altitudes of the level flagged as "z_opaque"
($Z^{|}_{Opaque}$) and of the highest ($Z^{|}_{Top}$) and lowest ($Z^{|}_{Base}$) levels flagged as "Cloud", using the temperature profiles of
the NASA Global Modeling and Assimilation Office (GMAO) reanalysis (Suarez et al., 2005) provided in CALIOP
Level 1 data and reported in GOCCP v3.0 data.
•   Thin cloud emissivity $\varepsilon^{|}_{Thin}$ of a *Thin cloud single column* is inferred from the mean attenuated scattering ratio of
levels flagged as "Clear" below the cloud, that we note $\langle SR' \rangle_{below}$ and which approximately corresponds to the
apparent two-way transmittance through the cloud. Indeed, considering a fixed multiple scattering factor $\eta = 0.6$,



we retrieve the Thin cloud visible optical depth $\delta_{Thin}^{VIS}$ (Garnier et al., 2015). Then, as the cloud particles are much
larger than the visible and infrared wavelengths and considering no absorption by cloud particles is occurring in the
visible domain, the Thin cloud LW optical depth $\delta_{Thin}^{LW}$ is approximately half of $\delta_{Thin}^{VIS}$ (Garnier et al., 2015). Finally,
we retrieve the Thin cloud emissivity with $\varepsilon_{Thin}^{|} = 1 - e^{-\delta_{Thin}^{LW}}$. Opaque cloud emissivity cannot be inferred and we
do the approximation that it is close to a black body, so $\varepsilon_{Opaque}^{|} \approx 1$.
Single columns with multi-layers of clouds are also consider in this study, i.e. $T_{Top}^{|}$ and $Z_{Top}^{|}$ refer to the highest
"Cloud" flagged level of the highest cloud in the column and $T_{Base}^{|}$ and $Z_{Base}^{|}$ to the lowest "Cloud" flagged level of the
lowest cloud in the column. Also, in this case, $\varepsilon_{Thin}^{|}$ is computed from the summed optical depth of all cloud layers present
in the column.
In order to avoid all possible uncertainties due to solar noise, results presented in this paper are only for nighttime
conditions. Furthermore, we restricted this study to observations over oceans to avoid uncertainties due to the ground
temperature diurnal cycle over land. And, in order to not be influenced by major changes of surface physical properties
across the seasons, we also removed from this study all observations over iced sea, based on sea ice fraction from the
European Centre for Medium-Range Weather Forecasts (ECMWF) ERA-Interim reanalysis (Berrisford et al., 2011).
**2.2 Fluxes observations collocated with lidar clouds observations**
CERES radiometer, on-board the Aqua satellite, measures the OLR at the same location where the CALIOP lidar,
on board the CALIPSO satellite, will shoot 2 minutes and 45 seconds afterwards. So, the instantaneous Single Scanner
Footprint (SSF) of the CERES swath crossing the CALIPSO ground-track give the OLR over atmospheric single columns
sounded by the lidar. Because a CERES footprint has a diameter of ~20 km, whereas the CALIOP lidar samples every 333 m
along-track with a footprint of 70 m diameter, several atmospheric single columns sounded by the lidar (up to 60) are located
within a single CERES footprint. To collocate the GOCCP-OPAQ instant data and the CERES SSF measurements, we use
the CALIPSO, CloudSat, CERES, and MODIS Merged Product (C3M; Kato et al., 2011) which flags the instantaneous
CERES SSF of the CERES swath crossing the CALIPSO ground-track. Finally, for each of these flagged CERES SSF, we
matched, from geolocation information, all the GOCCP-OPAQ single columns falling into the CERES footprint. We
consider that an atmospheric column with CERES footprint base is an Opaque (Thin) cloud column if all matched single
columns are declared as Opaque (Thin) cloud single column. We use these Opaque and Thin cloud columns to validate lidar-
derived OLR.
From the C3M product, we also use the estimated Clear sky OLR of the instantaneous CERES SSF of the CERES
swath crossing the CALIPSO ground-track. This estimated Clear sky OLR is computed from radiative transfer simulations
using the synergy information of the different instruments flying in the Afternoon Train (A-Train) satellite constellation. As
C3M is only released through April 2011, during the time period when both CALIPSO and CloudSat are healthy, we also
use the Clear sky OLR from 1°×1° gridded data monthly mean CERES Energy Balanced and Filled (EBAF) Edition 2.8
1°×1° product (Loeb et al., 2009), that we average over 2°×2° grid boxes.
**2.3 Radiative transfer computations**
For all radiative transfer computations needed in this study, we use the GAME radiative transfer code (Dubuisson et
al., 2004) combined with mean sea surface temperature (SST) and profiles of temperature, humidity and ozone extracted
from the ERA-Interim reanalysis. GAME is an accurate radiative transfer code to calculate the radiative flux and radiance
over the total solar and infrared spectrum. The radiative transfer equation is solved using DISORT (Stamnes et al., 1988) and
gaseous absorption is calculated from the k-distribution method. This code accounts for aerosol and clouds scattering and
absorption as well as interactions with gaseous absorption. GAME radiative transfer code does not take into account cloud



3D effects, and is based on the plane-parallel approximation. In this study, we use GAME to compute integrated OLR
between 5 and 100 μm.



**3 Radiative temperatures of Opaque clouds and Thin clouds derived from lidar cloud observations and reanalysis**
We define here an approximation of the Opaque and Thin cloud radiative temperatures which can be derived from
lidar measurements. The cloud radiative temperature corresponds to the equivalent radiative temperature of the cloud $T_{rad}^|$
such as the upward LW radiative flux emitted by a cloud of emissivity $\varepsilon^|$, at the top of the cloud, is $F_{Cloud}^{\uparrow LW|}(Cloud\ Top) =$
$\varepsilon^| \sigma (T_{rad}^|)^4$, where $\sigma$ denotes the Stefan–Boltzmann constant. We present distributions of these cloud radiative temperatures
derived from lidar measurements over the mid-latitudes region and the tropics.
**3.1 Definition and approximations of the cloud radiative temperature**
Considering an optically uniform cloud with a cloud total LW optical depth $\delta_{Cloud}^{LW|}$, and assuming a linear increase
of the temperature from the cloud top to the cloud base, the upward LW radiative flux emitted by the cloud at the top of the
cloud $F_{Cloud}^{\uparrow LW|}(Cloud\ Top)$ can be computed from the radiative transfer equation (RTE) (see appendix A). Then, solving the
equation $F_{Cloud}^{\uparrow LW|}(Cloud\ Top) = \varepsilon^| \sigma (T_{rad}^|)^4 = (1 - e^{-\delta_{Cloud}^{LW|}}) \sigma (T_{rad}^|)^4$, we can infer the value of the equivalent radiative
cloud temperature $T_{rad}^|$. Figure 3 shows $T_{rad}^|$ deduced from RTE (green) as a function of $\delta_{Cloud}^{LW|}$. As $\delta_{Cloud}^{LW|}$ increases, $T_{rad}^|$ is
found closer to the cloud top and so the cloud radiative temperature decreases.

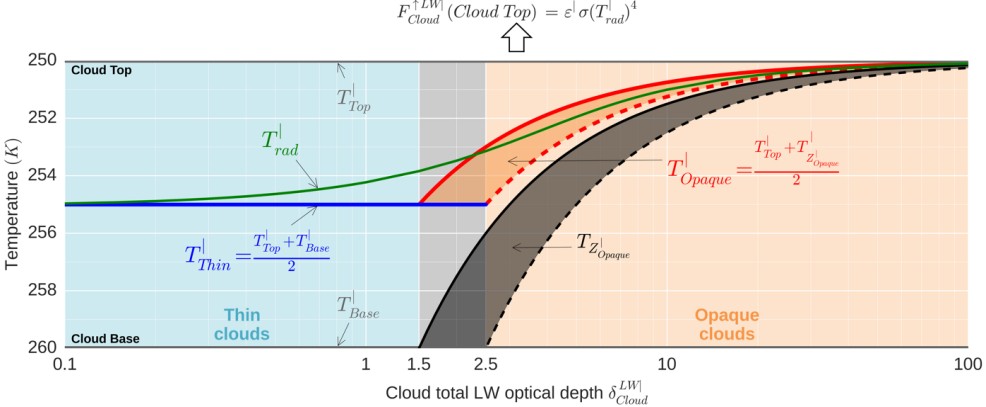

FIG. 3. Comparison of (green) the cloud radiative temperature $T_{rad}^|$ inferred from the RTE (see appendix A) with the lidar-definitions of (blue) the Thin cloud radiative temperature $T_{Thin}^|$ and (red) the Opaque cloud radiative temperature $T_{Opaque}^|$, as a function of the cloud total LW optical depth $\delta_{Cloud}^{LW|}$. Here, on an example with a fixed cloud top temperature $T_{Top}^|$ at 250 K and a fixed cloud base temperature $T_{Base}^|$ at 260 K.

$T_{rad}^|$ is obtained by computing the LW flux emitted by the cloud at the top of the cloud $F_{Cloud}^{\uparrow LW|}(Cloud\ Top)$ from the RTE and then
solving $F_{Cloud}^{\uparrow LW|}(Cloud\ Top) = \varepsilon^| \sigma (T_{rad}^|)^4$.

     Clouds are declared as (orange area) Opaque clouds if they present an opacity level altitude $Z_{Opaque}^|$. This occurs in lidar observations for $\delta_{Cloud}^{LW|}$ greater than a limit situated between 1.5 to 2.5. Below this limit clouds are declared as (blue area) Thin clouds. Clouds with $\delta_{Cloud}^{LW|}$ between 1.5 and 2.5 could be (gray area) either Opaque or Thin clouds depending on the limit.


We will now approximate $T_{rad}^|$ for Opaque clouds and Thin clouds using straightforward formulations which could
be derived from lidar cloud observations and reanalysis. In an Opaque cloud single column (Fig. 1, right), the optically very
thick cloud prevents LW radiative flux from below to propagate upwards. Thus, atmospheric layers below $Z_{Opaque}^|$ have
little influence on the OLR over an Opaque cloud single column $OLR_{Opaque}^|$. Here, we propose that $OLR_{Opaque}^|$ is mainly
driven by an *Opaque cloud radiative temperature* defined as:



$$T^|_{Opaque} = \frac{T^|_{Top} + T^|_{z^|_{Opaque}}}{2}.$$ (1)
In a Thin cloud single column (Fig. 1, left), the cloudy part is optically semi-transparent and lets through a part of the LW
radiative flux coming from the cloud-free atmospheric layers and surface underneath. Then, the OLR over a Thin cloud
single column $OLR^|_{Thin}$ depends on one hand on the surface temperature, the surface emissivity, the temperature profile, and
the humidity profile, and on the other hand on the cloud emissivity $\varepsilon^|_{Thin}$ and the *Thin cloud radiative temperature* defined
as:
$$T^|_{Thin} = \frac{T^|_{Top} + T^|_{Base}}{2}.$$ (2)

Comparisons of $T^|_{Thin}$, the cloud radiative temperature of Thin clouds ($\delta^{LW|}_{Cloud} < 1.5$, blue area), and $T^|_{Opaque}$, the

cloud radiative temperature of Opaque clouds ($\delta^{LW|}_{Cloud} > 2.5$, orange area), with $T^|_{rad}$ deduced from RTE (green) show good
agreement in Fig. 3. Clouds with $1.5 < \delta^{LW|}_{Cloud} < 2.5$ (gray area) can be either Thin or Opaque clouds depending on the
integrated LW optical depth at which $Z^|_{Opaque}$ will occur. Here, computations were performed for a fixed cloud top
temperature $T^|_{Top}$ at 250 K and a fixed cloud base temperature $T^|_{Base}$ at 260 K. $T^|_{Opaque}$ will depend on the integrated LW
optical depth from cloud top $\delta^{LW|}$ to where $Z^|_{Opaque}$ will occur, which is known to be situated between 1.5 and 2.5: $\delta^{LW|} =$
$\frac{1}{2}\delta^{VIS|}$ (Chepfer et al., 2014), with $\delta^{VIS|}$ between 3 and 5 (Vaughan et al., 2009). Then, according to $Z^|_{Opaque}$ possible values
given this approximation (black shadow area), $T^|_{Opaque}$ range is deduced (red shadow area).

Computations with other pairs of $T^|_{Top}$ and $T^|_{Base}$ temperatures (not shown) reveal that the relative vertical position

into the cloud of $T^|_{rad}$ does not depend much of the cloud top and cloud base temperatures. In other words, with other pairs
of $T^|_{Top}$ and $T^|_{Base}$ temperatures, we obtain almost the same figure as Fig. 3 only with the y-axis temperature values changed.
This means that the difference between $T^|_{rad}$ and $T^|_{Thin}$ or between $T^|_{rad}$ and $T^|_{Opaque}$ becomes larger as the difference
between $T^|_{Top}$ and $T^|_{Base}$ increases. Naturally, in reality, the error made by using $T^|_{Thin}$ and $T^|_{Opaque}$ as approximations of $T^|_{rad}$
will also depend on other cloud properties, such as cloud inhomogeneity and cloud microphysics. However, this simple
theoretical calculation allows us to assert that $T^|_{Thin}$ and $T^|_{Opaque}$ as we defined above are good approximations of the cloud
radiative temperature of the Thin and Opaque clouds, with less than a 2 K error for a Thin cloud with a 10 K difference
between its cloud base and cloud top temperatures, and less than a 1 K error for an Opaque cloud with $\delta^{LW|}_{Cloud} > 5$ and with a
10 K difference between its cloud base and cloud top temperatures.
**3.2 $T^|_{Opaque}$ and $T^|_{Thin}$ retrieved from CALIOP observations during 2008–2015**

For each cloudy single column sounded by CALIOP, we derive $T^|_{Opaque}$ from $T^|_{Top}$ and $T^|_{z^|_{Opaque}}$ using Eq. (1), and

we derive $T^|_{Thin}$ from $T^|_{Top}$ and $T^|_{Base}$ using Eq. (2). Then, we computed the probability density function (PDF) of $T^|_{Opaque}$
among Opaque clouds and $T^|_{Thin}$ among Thin clouds for 3 different regions: the tropical ascending region between
$\pm 30°$ latitude with monthly mean 500-hPa pressure velocity $\omega_{500} < 0$ hPa·day⁻¹, the tropical subsiding region between
$\pm 30°$ latitude with monthly mean $\omega_{500} > 0$ hPa·day⁻¹ and the mid-latitudes (North and South) region between 65° S and
30° S and between 30° N and 65° N put together. To compute these PDFs, e.g. the PDF of $T^|_{Opaque}$ among Opaque clouds,
we firstly compute a PDF of $T^|_{Opaque}$ among all single columns on each 2°×2° grid box for the 2008–2015 period. Then, we
compute the area-weighted averaged PDF of a region, weighting each 2°×2° grid box PDF by the ratio of the number of





Opaque single columns over the number of all single columns. We do this in order to take into account the sampling differences in each grid box.

Figure 4a shows the distributions of $T^|_{Opaque}$ among Opaque clouds. In the tropical subsiding region (green), 71 % of $T^|_{Opaque}$ are found between 0 °C and 25 °C with a maximum at 15 °C. Because they are warm, they do not strongly affect the OLR compared to clear-sky conditions. These clouds are the marine boundary layer clouds of the descending branches of the Hadley cells. In the tropical ascending region (red), $T^|_{Opaque}$ has a bimodal distribution with few warm clouds between 0 °C and 25 °C (21 %) and most cloud temperatures spread between 0 °C and -80 °C (79 %). These latter Opaque clouds will have locally a very strong impact on the OLR since their temperatures are up to 100 K lower than the surface. However, the tropical ascending region only represents about 1/5 of the ocean surface between 65° S and 65° N making their effect at global scale less striking. In the mid-latitudes region (purple), $T^|_{Opaque}$ are unsurprisingly located at temperatures less extreme than in tropical regions with temperatures ranging from 20 °C to -60 °C and are rather evenly distributed between 10 °C and -30 °C. These Opaque clouds will have a mid-effect on the local OLR, but the mid-latitudes region represent a large area (43 % of the ocean surface between 65° S and 65° N) and their cover over these regions is large (Fig. 2a). So, they will certainly also play an important role on the global CRE.

The Opaque cloud radiative temperature $T^|_{Opaque}$ is based on the key new lidar information $Z^|_{Opaque}$ (Eq. (1)). Figure 4b shows that $Z^|_{Opaque}$ is mostly low in all regions, at around 1 km altitude, in the boundary layer clouds, especially for the subsiding region. Some non-negligible amount of $Z^|_{Opaque}$ are found between 2 km and 8 km in the mid-latitudes storm tracks. In the tropical ascending region, the PDF is tri-modal with a first pick around 1 km, a second around 5 km and a third around 12 km, suggesting the presence of Opaque clouds in the boundary layer and at very high altitudes due to deep convection for the first and last mode. The second mode could be due to more diffuse or developing convective clouds. Since $T^|_{Opaque}$ also depends on $Z^|_{Top}$, distributions of the distance between cloud top and $Z^|_{Opaque}$ among Opaque clouds are given in Fig. A1a (appendix B).

As in Fig. 4a but for Thin clouds, Fig. 4c also shows, in the tropical subsiding region, a large majority of $T^|_{Thin}$ higher than 0 °C (65 %). $T^|_{Thin}$ colder than -40 °C are more frequent than for $T^|_{Opaque}$, suggesting high-altitude optically thin cirrus from detrainments of anvil clouds generated in adjacent convective regions. In the tropical ascending region, the "warm" mode of the bimodal distributions of $T^|_{Thin}$ is bigger and warmer than that of $T^|_{Opaque}$. The main mode of $T^|_{Thin}$ in the mid-latitudes region, is also warmer than that of $T^|_{Opaque}$. Warmer cloud temperatures, implying smaller CRE, reinforces the importance of the role of the Opaque clouds versus the Thin clouds in the total CRE. Distributions of the distance between cloud top and cloud base among Thin clouds are given in Fig. A1b (appendix B).

Because the radiative impact of the Thin clouds will also depend on the cloud emissivity of the cloud, we also computed the distributions of $\varepsilon^|_{Thin}$ among Thin clouds. Figure 4d shows these distributions. For all regions, the maximum is located around 0.25. So, emissivities of Thin clouds are usually small, and clouds with small emissivities have less impact on the OLR. This, once again, goes in the sense that the role that play Thin clouds on the total CRE should be significantly smaller than that of Opaque clouds.





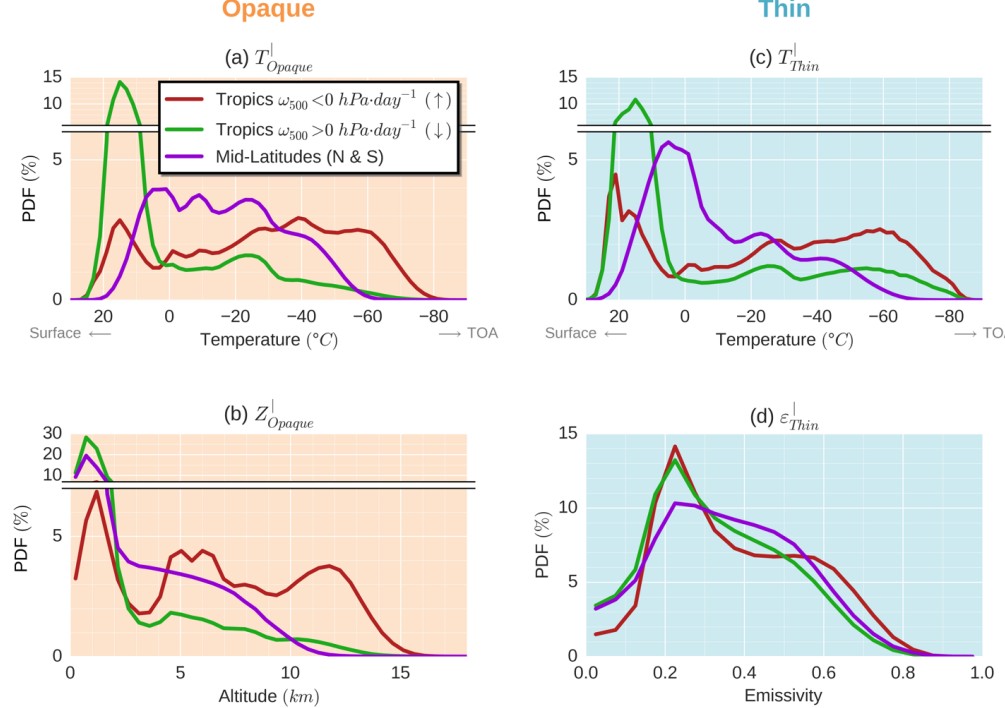


FIG. 4. Observed distributions of (a) $T^|_{Opaque}$ among Opaque clouds, (b) $Z^|_{Opaque}$ among Opaque clouds, (c) $T^|_{Thin}$ among Thin clouds
and (d) $\varepsilon^|_{Thin}$ among Thin clouds in three regions: (red) the tropical [30° S–30° N] ascending regime areas (monthly mean $\omega_{500} < 0$
hPa·day⁻¹), (green) the tropical [30° S–30° N] subsiding regime areas (monthly mean $\omega_{500} > 0$ hPa·day⁻¹) and (purple) the mid-latitudes
[30°–65°]. These regions represent respectively 22 %, 35 % and 43 % of their total area. Only nighttime over ice-free oceans for the 2008-
2015 period is considered.





**4. Outgoing longwave radiation derived from lidar cloud observations**
In this section, we express the OLR as a function of cloud properties derived from lidar observations ($T_{Opaque}^{|}$,
$T_{Thin}^{|}$, and $\varepsilon_{Thin}^{|}$). Then, we verify this relationship against observations at instantaneous 20 km footprint scale, using high
spatial resolution collocated satellite-borne broadband radiometer and lidar data, and at monthly mean
2° latitude × 2° longitude gridded scale.
**4.1 Linear relationship deduced from radiative transfer simulations over single cloudy column**
The goal of this sub-section is to establish a simple and robust relationship between 1) the OLR over an Opaque
cloud single column $OLR_{Opaque}^{|}$ and the radiative temperature $T_{Opaque}^{|}$ and, 2) the OLR over a Thin cloud single column
$OLR_{Thin}^{|}$ and the radiative temperature $T_{Thin}^{|}$ and the Thin cloud emissivity $\varepsilon_{Thin}^{|}$.
1) For an Opaque cloud single column, we computed $OLR_{Opaque}^{|}$, using direct radiative transfer computations, for
various atmospheres containing an Opaque cloud with different altitudes and vertical extents, represented by a cloud layer
with emissivity equal to 1 at $Z_{Opaque}^{|}$ topped with optically uniform cloud layers with vertically integrated visible optical
depth equal to 3.2, which correspond to $\varepsilon \approx 0.8$. Figure 5a shows on dots the obtained $OLR_{Opaque}^{|}$ as function of $T_{Opaque}^{|}$ for
tropical atmosphere conditions. Linear regression (solid line) leads to:
$$OLR_{Opaque}^{|\,(LID)} = 2.0 T_{Opaque}^{|} - 310. \tag{3}$$

where $OLR_{Opaque}^{|\,(LID)}$ is expressed in W·m$^{-2}$ and $T_{Opaque}^{|}$ in K. So, when $T_{Opaque}^{|}$ decreases of 1 K (e.g. if the Opaque cloud
rises up) then the OLR decreases by 2 W·m$^{-2}$. This linear relationship, firstly found by Ramanathan (1977), has a slope
which is consistent with previous work that found 2.24 W·m$^{-2}$/K (Wang et al. (2002) using the radiative transfer model of Fu
and Liou (1992, 1993) and the analysis of Kiehl (1994)). Linear regressions done on other regions with different atmospheric
conditions give a similar coefficient. This means that, in spite of the significant differences in the atmospheric temperature
and humidity profiles, $OLR_{Opaque}^{|}$ depends essentially only on $T_{Opaque}^{|}$. This remarkable result demonstrates a cloud property
which drives the OLR can be derived from spaceborne lidar measurement. Figure 5a also shows the black body emission
(dashed line). Differences between the computed OLR and the black body emission represent the extinction effect of the
atmospheric layers located above the cloud.
2) For a Thin cloud single column, we can consider $OLR_{Thin}^{|}$ composed of two parts (Fig. 1). A first part, coming
from the LW flux emitted by the cloud, which can be expressed in the same way as Eq. (3) using $T_{Thin}^{|}$ instead of $T_{Opaque}^{|}$,
and weighted by the Thin cloud emissivity $\varepsilon_{Thin}^{|}$. The second part is equal to the OLR over a Clear sky single column
$OLR_{Clear}^{|}$ (the same single column without the cloud) multiplied by the cloud transmissivity $(1 - \varepsilon_{Thin}^{|})$:
$$OLR_{Thin}^{|\,(LID)} = \varepsilon_{Thin}^{|}(2.0 T_{Thin}^{|} - 310) + (1 - \varepsilon_{Thin}^{|})OLR_{Clear}^{|}. \tag{4}$$

where $OLR_{Thin}^{|\,(LID)}$ and $OLR_{Clear}^{|}$ are expressed in W·m$^{-2}$ and $T_{Thin}^{|}$ in K. In order to evaluate this expression and to examine
the dependence of $OLR_{Thin}^{|}$ to $T_{Thin}^{|}$ and $\varepsilon_{Thin}^{|}$, we computed $OLR_{Thin}^{|}$, using direct radiative transfer computations, for
various atmospheres containing a Thin cloud (represented by optically uniform cloud layers with integrated emissivities
equal to $\varepsilon_{Thin}^{|}$) with different altitudes, vertical extents and emissivities. Figure 5b shows on dots the resulting $OLR_{Thin}^{|}$ as
function of $T_{Thin}^{|}$ for 4 different values of $\varepsilon_{Thin}^{|}$, for tropical atmosphere conditions. We compare these results with the linear
expression of Eq. (4) (solid lines), in which $OLR_{Clear}^{|}$ is obtained by computing the OLR for a single column without cloud.
The theoretical formulation agrees quite well with the different simulations. It may be noted, however, that this formulation
seems to overestimate $OLR_{Thin}^{|}$ (up to +10 W·m$^{-2}$) for many cases. Reasons for it are discussed in Section 6.






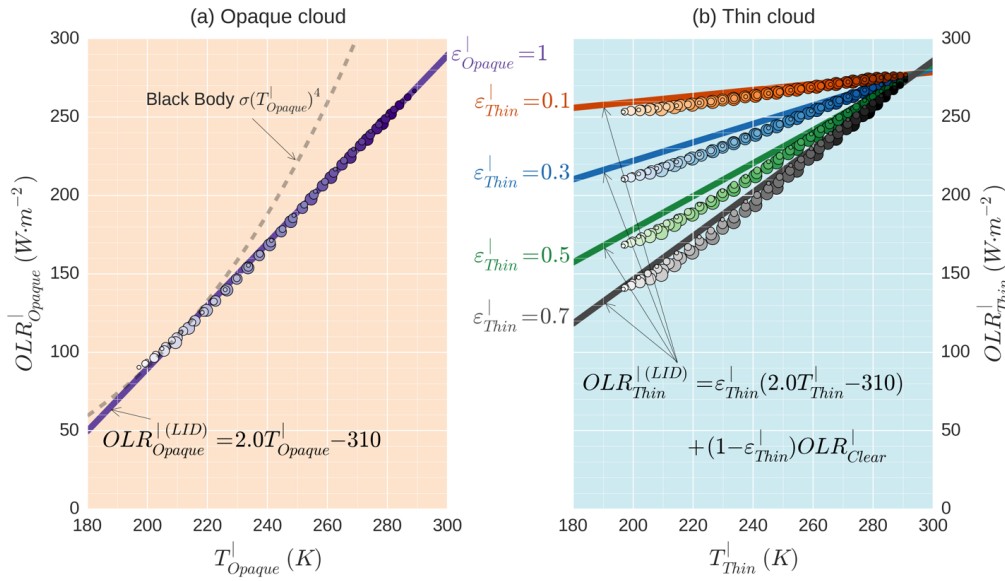


FIG. 5. Relationship between the OLR and the cloud radiative temperature from radiative transfer computations: (a) over an Opaque cloud single column and (b) over a Thin cloud single column. Direct radiative transfer computations are shown in dots. Solid lines represent the linear relationships inferred from a regression on dots in the Opaque case and applied to the Thin clouds case according to Eq. (4). For a fixed value of cloud emissivity (dots colors; 1 [purples] for Opaque clouds and 0.1 [reds], 0.3 [blues], 0.5 [greens], 0.7 [greys] for Thin clouds), the linear relationship does not depend on the cloud altitudes (dots light intensity; 0 km [dark] – 16 km [bright]) or the geometrical thicknesses (dots size; 1 km [small] – 5 km [large]). Results shown here use the 2008-year mean thermodynamic atmospheric variables over the tropical region [30° S–30° N] from ERA-I reanalysis.


**4.2 Evaluation of the linear relationship using observations at instantaneous CERES footprint scale**


We evaluate the robustness of the OLR expressions (Eqs. (3) and (4)) at the resolution of a CERES footprint

(~20 km) using CERES measurements, and cloud properties derived from collocated CALIOP observations $T_{Opaque}^{\oslash}, T_{Thin}^{\oslash}$
and $\varepsilon_{Thin}^{\oslash}$. For this purpose, we apply Eqs. (3) and (4) using $T_{Opaque}^{\oslash}, T_{Thin}^{\oslash}, \varepsilon_{Thin}^{\oslash}$ and     the estimated OLR over the scene
removing the clouds given by C3M $OLR_{Clear}^{\oslash}$. $T_{Opaque}^{\oslash}, T_{Thin}^{\oslash}, \varepsilon_{Thin}^{\oslash}$ refer    to    atmospheric column with a CERES footprint
base, as mentioned by the exponent symbol "$\oslash$", and are   obtained   averaging   respectively    all $T_{Opaque}^{|}, T_{Thin}^{|}$ and $\varepsilon_{Thin}^{|}$ falling
into the CERES footprint. $OLR_{Clear}^{\oslash}$, $OLR_{Opaque}^{\oslash}$ and $OLR_{Thin}^{\oslash}$ refer to atmospheric column with a CERES footprint base.

Figure 6a compares the $OLR_{Opaque}^{\oslash\,(CERES)}$ measured   by   CERES   only   over footprints entirely cover by an Opaque cloud,

with the $OLR_{Opaque}^{\oslash\,(LID)}$ computed from $T_{Opaque}^{\oslash}$ using Eq. (3). We see a very strong correlation between observed and computed
OLR (R = 0.95). Therefore, this confirms that the OLR over an Opaque cloud is linearly dependent of $T_{Opaque}^{\oslash}$. So, from
lidar measurement it is possible to derive a cloud property which is proportional to the OLR. Monitoring $T_{Opaque}^{|}$ on long-
term should provide important information which should help to better understand the LW cloud feedback mechanism.
Moreover, because the relationship is linear, it simplifies the derivatives in mathematical expressions of feedback and will
allow to construct a useful framework to study LW cloud feedback in simulations of climate models.

Figure 6b is the same as Fig. 6a but only for CERES footprints entirely cover by a Thin cloud. So $OLR_{Thin}^{\oslash\,(LID)}$ is

computed from $T_{Thin}^{\oslash}, \varepsilon_{Thin}^{\oslash}$ and $OLR_{Clear}^{\oslash}$ using Eq. (4). $OLR_{Thin}^{\oslash\,(LID)}$ compared to observations ($OLR_{Thin}^{\oslash\,(CERES)}$) also shows
quite good correlation (R = 0.89), but the regression slightly differs from the identity line. Possible reasons for disagreements



between observed $OLR_{Thin}^{\oslash\,(LID)}$ and computed $OLR_{Thin}^{\oslash\,(CERES)}$ are discussed in Section 6. These same results are also drawn as
function of $T_{Thin}^{\oslash}$ and $\varepsilon_{Thin}^{\oslash}$ in Fig. A2 for a fixed value of $OLR_{Clear}^{\oslash}$ (we selected measurements where $OLR_{Clear}^{\oslash} \in$
$[275, 285]$ W·m$^{-2}$) in order to show the effect of those two cloud properties on $OLR_{Thin}^{\oslash\,(CERES)}$.
The evaluation showed in Fig. 6 is only using observation from January 2008. The same evaluation performed with
July 2008 data (not shown) gives similar results, with R = 0.96 for Opaque clouds and R = 0.90 for Thin clouds.

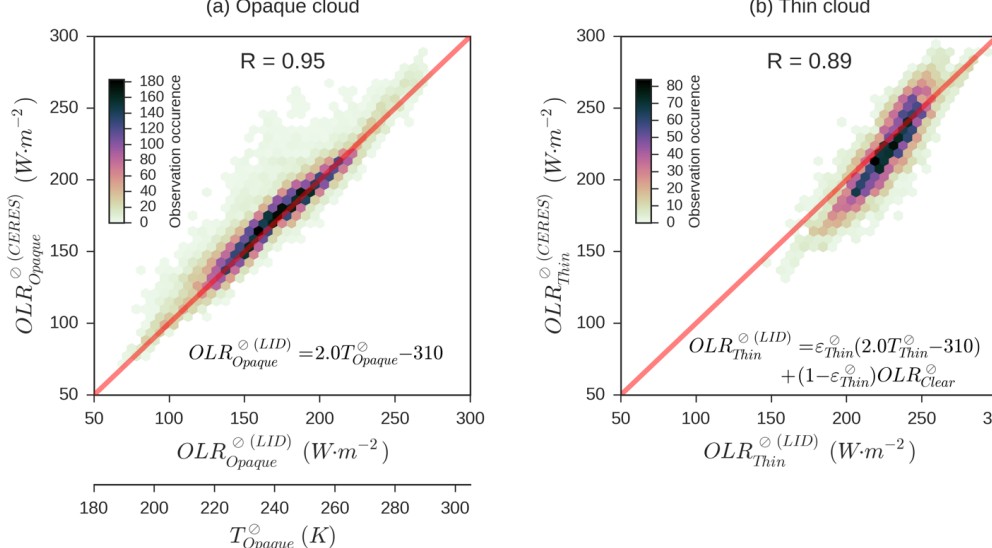

Fig. 6. Comparison between observed and lidar-derived OLR at CERES footprint scale: (a) over Opaque cloud single columns and (b) over Thin cloud single columns. Results obtained from CERES (y-axis) and CALIOP (x-axis) collocated measurements. $OLR_{Opaque}^{\oslash\,(LID)}$ and $OLR_{Thin}^{\oslash\,(LID)}$ are computed using Eqs. (4) and (5). Only nighttime over ice-free oceans for January 2008 is considered. $R$ is the correlation coefficient.


**4.3 Evaluation of the linear relationship using observations at monthly mean 2°×2° gridded scale**

We first compute the monthly mean gridded total OLR from gridded lidar cloud properties:
$$OLR_{Total}^{\boxplus\,(LID)} = C_{Clear}^{\boxplus} OLR_{Clear}^{\boxplus} + C_{Opaque}^{\boxplus} OLR_{Opaque}^{\boxplus\,(LID)} + C_{Thin}^{\boxplus} OLR_{Thin}^{\boxplus\,(LID)}, \qquad (5)$$
where $C_{Clear}^{\boxplus}$, $C_{Opaque}^{\boxplus}$ and $C_{Thin}^{\boxplus}$ are the monthly mean covers (Figs. 1,2): the ratio between the number of a specific
kind of single column over the total number of single columns that fall into the grid box during a month. $OLR_{Opaque}^{\boxplus\,(LID)}$ is
computed from $T_{Opaque}^{\boxplus}$ using Eq. (3), and $OLR_{Thin}^{\boxplus\,(LID)}$ is computed from $T_{Thin}^{\boxplus}$, $\varepsilon_{Thin}^{\boxplus}$ and $OLR_{Clear}^{\boxplus}$ using Eq. (4). $T_{Opaque}^{\boxplus}$,
$T_{Thin}^{\boxplus}$ and $\varepsilon_{Thin}^{\boxplus}$ are obtained by averaging respectively all $T_{Opaque}^{|}$, $T_{Thin}^{|}$ and $\varepsilon_{Thin}^{|}$ falling into the 2°×2° box.
We then evaluate the lidar-derived $OLR_{Total}^{\boxplus\,(LID)}$ against the CERES measurements $OLR_{Total}^{\boxplus\,(CERES)}$. To do so, we
computed the 2008–2010 mean $OLR_{Total}^{\boxplus\,(LID)}$ from Eq. (5) using $OLR_{Clear}^{\boxplus}$ from C3M and compared it with the one measured
by CERES-Aqua. Figure 7 shows the comparison between computed $OLR_{Total}^{\boxplus\,(LID)}$ (Fig. 7a) and measured $OLR_{Total}^{\boxplus\,(CERES)}$
(Fig. 7b). We firstly observe the noteworthy agreement of OLR patterns. Figure 7c shows the difference between those two
maps. The global mean difference is -0.1 W·m$^{-2}$, meaning $OLR_{Total}^{\boxplus\,(LID)}$ very slightly underestimate the observed
$OLR_{Total}^{\boxplus\,(CERES)}$. The zonal mean differences (not shown) are quite small and never exceed 5 W·m$^{-2}$ and are mostly lower than





2 W·m⁻². Locally, we note a lack of OLR over the warm pool, the Intertropical Convergence Zone (ITCZ) and the
stratocumulus regions off the West coast of continents (up to 6–8 W·m⁻²) and an excess of OLR over latitudes beyond 50° N
or 40° S (up to 4–6 W·m⁻²). As C3M only covers through April 2011, but we aim to use this framework on long time-series
observations, we replace $OLR^{\boxplus}_{Clear}$ from C3M by $OLR^{\boxplus}_{Clear}$ from CERES-EBAF in the following of this paper. Comparison
between observed and lidar-derived OLR using $OLR^{\boxplus}_{Clear}$ from CERES-EBAF instead of $OLR^{\boxplus}_{Clear}$ from C3M is showed in
Fig. A3. Using $OLR^{\boxplus}_{Clear}$ from C3M increases the global mean $OLR^{\boxplus (LID)}_{Total}$ by 0.6 W·m⁻². Reasons for this increase are
discussed in Section 6.

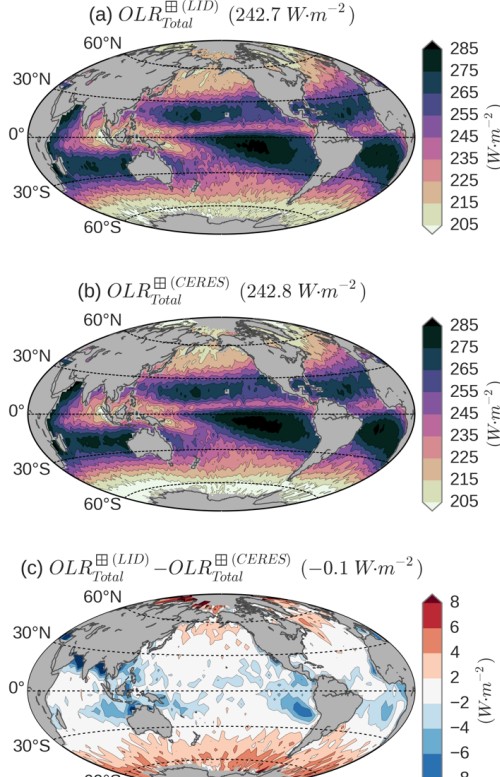

FIG. 7. Comparison between observed and lidar-derived OLR at 2°×2° gridded scale: (a) derived from CALIOP observations and (b) measured by CERES-Aqua. (c) = (a) - (b). Only from nighttime over ice-free oceans for the 2008–2010 period is considered. Global mean values are given in parentheses.





**5 Contributions of Opaque clouds and Thin clouds to the cloud radiative effect**


In the previous section, we found a clear linear relationship for Opaque clouds between $OLR_{Opaque}$ and $T_{Opaque}$ at


different scales. The relationship for Thin clouds, though quite simple, is not linear and agrees less with observations than for


Opaque clouds. In this section, we evaluate the contributions of Opaque clouds and Thin clouds to the total CRE.


**5.1 Partitioning cloud radiative effect into Opaque CRE and Thin CRE**


Using Eq. (5), we are able to decompose the total CRE at the TOA, computed from lidar observations, in its Opaque


and Thin clouds contributions:


$$CRE_{Total}^{\boxplus\,(LID)} = OLR_{Clear}^{\boxplus} - OLR_{Total}^{\boxplus\,(LID)}$$


$$= C_{Opaque}^{\boxplus}\left(OLR_{Clear}^{\boxplus} - OLR_{Opaque}^{\boxplus\,(LID)}\right) + C_{Thin}^{\boxplus}\left(OLR_{Clear}^{\boxplus} - OLR_{Thin}^{\boxplus\,(LID)}\right). \quad (6)$$


$$\underbrace{\hspace{5cm}}_{CRE_{Opaque}^{\boxplus\,(LID)}} \qquad \underbrace{\hspace{5cm}}_{CRE_{Thin}^{\boxplus\,(LID)}}$$



Thereby, using Eq. (3), we can express $CRE_{Opaque}^{\boxplus\,(LID)}$ as a function of $C_{Opaque}^{\boxplus}$, $T_{Opaque}^{\boxplus}$ and $OLR_{Clear}^{\boxplus}$:


$$CRE_{Opaque}^{\boxplus\,(LID)} = C_{Opaque}^{\boxplus}\left(OLR_{Clear}^{\boxplus} - 2.0T_{Opaque}^{\boxplus} + 310\right). \quad (7)$$


Using Eq. (4), we can express $CRE_{Thin}^{\boxplus\,(LID)}$ as a function of $C_{Thin}^{\boxplus}$, $T_{Thin}^{\boxplus}$, $\varepsilon_{Thin}^{\boxplus}$ and $OLR_{Clear}^{\boxplus}$:


$$CRE_{Thin}^{\boxplus\,(LID)} = C_{Thin}^{\boxplus}\varepsilon_{Thin}^{\boxplus}\left(OLR_{Clear}^{\boxplus} - 2.0T_{Thin}^{\boxplus} + 310\right). \quad (8)$$


**5.2 Global means of the Opaque cloud CRE and the Thin cloud CRE**


Figure 8 shows the zonal mean observations of the 5 cloud properties ($C_{Opaque}^{\boxplus}$, $T_{Opaque}^{\boxplus}$, $C_{Thin}^{\boxplus}$, $T_{Thin}^{\boxplus}$ and $\varepsilon_{Thin}^{\boxplus}$). In


the subsidence branches of the Hadley cell, around 20° S and 20° N, $C_{Opaque}^{\boxplus}$ is minimum (Fig. 8a), $T_{Opaque}^{\boxplus}$ and $T_{Thin}^{\boxplus}$ are


warm (Fig 8b, temperatures in y-axis oriented downward) and $\varepsilon_{Thin}^{\boxplus}$ is minimum (Fig. 8c). So, we do not expect a very large


contribution to the CRE from these regions. In contrast, the Intertropical Convergence Zone (ITCZ) corresponds to local


maxima of Opaque and Thin cloud covers, extremely cold $T_{Opaque}^{\boxplus}$ and $T_{Thin}^{\boxplus}$ and a maximum of $\varepsilon_{Thin}^{\boxplus}$. Very large CRE will


arise from there. Interestingly, an inversion of cover predominance and colder temperature between Opaque and Thin clouds


occurs around 30° latitude. This suggests that the relative contribution of the Thin clouds to the CRE is larger in the tropical


belt than in the rest of the globe. This should not be very dependent on the year since the interannual variations of these 5


cloud properties (represented by the shaded areas) are very small compared to the zonal differences.







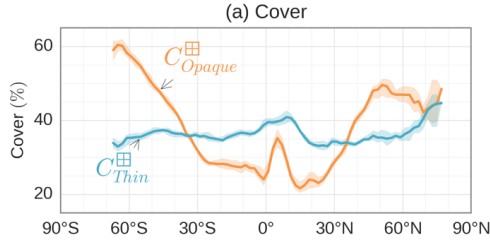

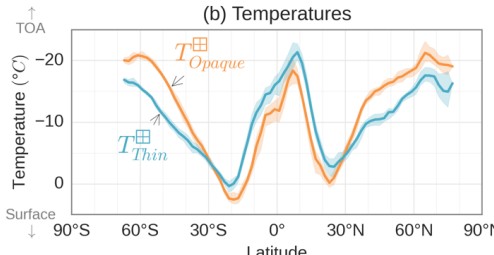

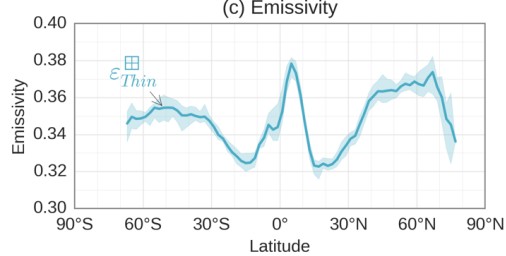


FIG. 8. Zonal mean observations: (a) $C_{Opaque}^{\boxplus}$ and $C_{Thin}^{\boxplus}$, (b) $T_{Opaque}^{\boxplus}$ among Opaque clouds and $T_{Thin}^{\boxplus}$ among Thin clouds and (c) $\varepsilon_{Thin}^{\boxplus}$ among Thin clouds. Only nighttime over ice-free oceans for the 2008–2015 period is considered. Shaded areas represent the envelope including interannual variations.


Figure 9 shows that Opaque clouds contribute the most (73 %) to the total CRE. We can also note that the zonal

variations of $CRE_{Opaque}^{\boxplus\,(LID)}$, and so approximately the variations of $CRE_{Total}^{\boxplus\,(LID)}$ (black line), can be explained by the zonal
variations of $T_{Opaque}^{\boxplus}$ and $C_{Opaque}^{\boxplus}$ (Fig. 8a,b). For example, the absolute maximum $CRE$ at 5° N (~44 W·m⁻²) is associated
with a large cover and cold temperature of Opaque clouds. As suggested hereinbefore, we see that the relative contribution
of Thin clouds ($CRE_{Thin}^{\boxplus\,(LID)}/CRE_{Total}^{\boxplus\,(LID)}$, Fig. 9b) is larger under the tropics, approximately 2 times larger below 30° (up to
40 %) than beyond those latitudes.





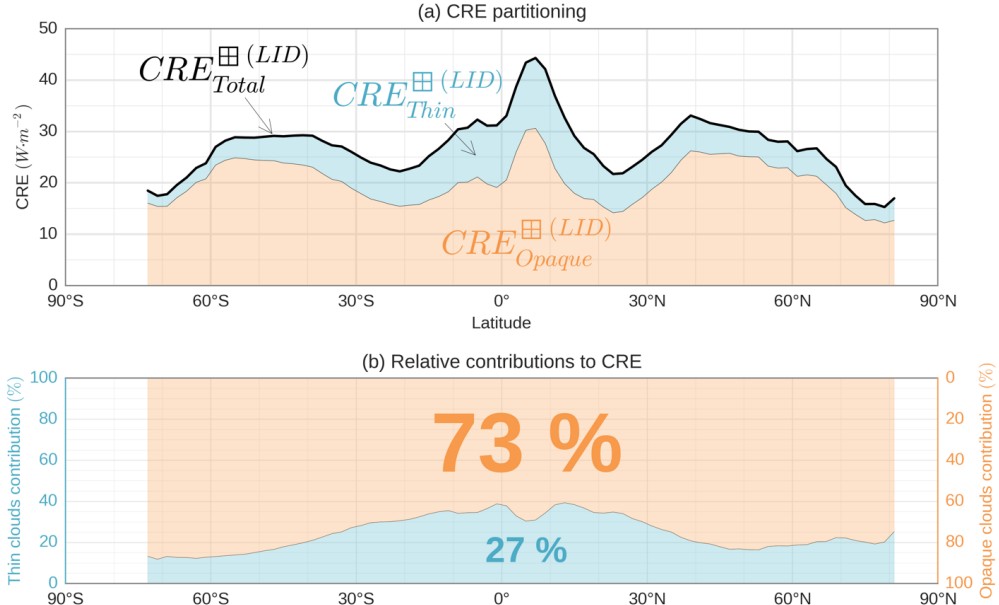

FIG. 9. (a) Partitioning of total CRE into Opaque CRE and Thin CRE. (b) Ratios of the Opaque (Thin) CRE to the total CRE. Only nighttime over ice-free oceans for the 2008–2015 period is considered.

Figure 10 shows the same CRE partitioning on maps. The likeness of patterns between total CRE (Fig. 10a) and the Opaque clouds CRE contribution (Fig. 10b) is prominent, strengthening again the importance of the Opaque clouds in the CRE. We can also note that Thin clouds CRE contribution (Fig. 10c) have quite large values between 20° S and 20° N in the Indian Ocean and the West Pacific Ocean, especially all around Indonesia, where $C_{Thin}^{\boxplus}$ (Fig. 2b) is maximum and $T_{Thin}^{\boxplus}$ minimum (not shown).



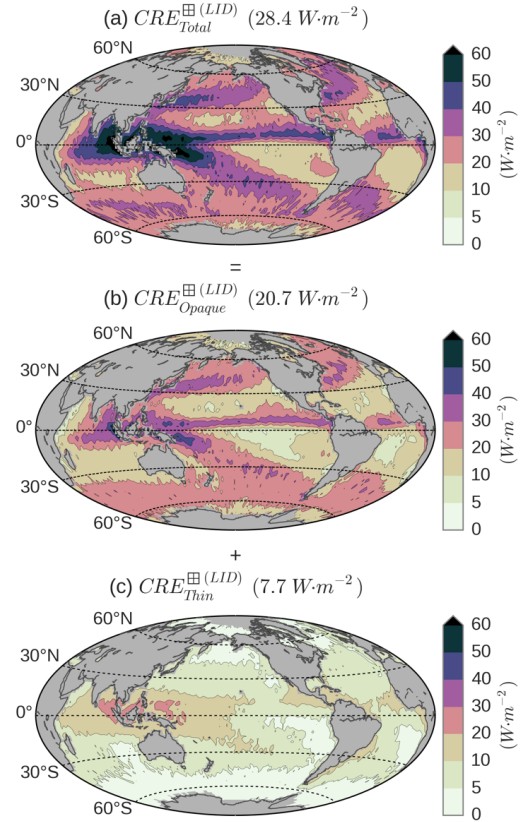

FIG. 10. Maps of (a) the total CRE (b) the Opaque CRE
and (c) the Thin CRE. Only nighttime over ice-free oceans
for the 2008–2015 period is considered. Global mean values
are given in parentheses.

Globally, the predominance of $CRE_{Opaque}^{\boxplus (LID)}$ is obvious since it represents nearly the three-fourth of the total

$CRE_{Total}^{\boxplus (LID)}$. Thereby, the cloud property $T_{Opaque}^{\boxplus}$ inferred from lidar observations and linearly linked to $OLR_{Opaque}^{\boxplus}$ should

be a very good candidate to constrain LW cloud feedbacks since Thin clouds only account for 27 % of $CRE_{Total}^{\boxplus (LID)}$. Also,

since the expression used for Thin clouds seems to give coherent results for $CRE_{Thin}^{\boxplus (LID)}$, it could also be used in a future

work to quantify the role of a change in $C_{Thin}^{\boxplus}$, $T_{Thin}^{\boxplus}$, and $\varepsilon_{Thin}^{\boxplus}$ in the variations of $CRE_{Thin}^{\boxplus (LID)}$.

**5.3 Tropical Opaque cloud CRE and Thin cloud CRE in dynamical regimes**

Figure 11 shows the cloud properties as a function of dynamical regime in the tropics (whose PDF according to the

500-hPa pressure velocity is given Fig. 11h). In the tropical convective regimes ($\omega_{500} < 0$ hPa·day$^{-1}$), $C_{Opaque}^{\boxplus}$ is strongly

driven (25 % to 45 % increase from 0 hPa·day$^{-1}$ to -100 hPa·day$^{-1}$) by the velocity of ascending air, whereas $C_{Thin}^{\boxplus}$ seems to

be poorly dependent of it, with an almost constant cover around 40 %. In subsidence regions, the mean $C_{Opaque}^{\boxplus}$ is also

increasing when the air descending velocity is larger but with a wide range of variation from month to month (Fig. 11a).

More strikingly, $T_{Opaque}^{\boxplus}$ and $T_{Thin}^{\boxplus}$ (Fig. 11b) vary linearly with $\omega_{500}$, with a small variability from month to month. $T_{Opaque}^{\boxplus}$



and $T_{Thin}^{\boxplus}$ linearly decrease from 20 hPa·day$^{-1}$ to -100 hPa·day$^{-1}$ from approximately 5 °C to -35 °C and are constant between
20 hPa·day$^{-1}$ and 70 hPa·day$^{-1}$ at 5 °C. This suggests that, locally, $T_{Opaque}^{\boxplus}$ and $T_{Thin}^{\boxplus}$ are invariants in each dynamical
regime. Radiative cloud temperatures $T_{Opaque}^{\boxplus}$ and $T_{Thin}^{\boxplus}$ presented in Fig. 11b were built respectively from temperatures at
altitudes $Z_{Opaque}^{|}$ and $Z_{Top}^{|}$, and from temperatures at altitudes $Z_{Base}^{|}$ and $Z_{Top}^{|}$ (see Section 3.1). The linear decrease from
20 hPa·day$^{-1}$ to -100 hPa·day$^{-1}$ of $T_{Opaque}^{\boxplus}$ and $T_{Thin}^{\boxplus}$ is due to the cumulative effects of a rising of the altitude of "apparent
cloud base" ($Z_{Opaque}^{|}$ for Opaque clouds and $Z_{Base}^{|}$ for Thin clouds; see monthly mean 2°×2° gridded $Z_{Opaque}^{\boxplus}$ and $Z_{Thin}^{\boxplus}$ on
Fig. 11c) and an elongation of the cloud vertical distribution which gives even higher $Z_{Top}^{|}$ (see monthly mean 2°×2° gridded
distance of "apparent cloud base" $Z_{Top}^{\boxplus} - Z_{Opaque}^{\boxplus}$ and $Z_{Top}^{\boxplus} - Z_{Base}^{\boxplus}$ on Fig. 11d). Figure 11e shows the distribution in
dynamical regimes of $\varepsilon_{Thin}$. It increases from 0.31 to 0.42 between 20 hPa·day$^{-1}$ and -100 hPa·day$^{-1}$, being almost invariant
from month to month, and it is around 0.32 in average in subsidence region.
An interesting point which appears in these figures is, in the tropics, the very small variability in the relationship
between cloud properties and $\omega_{500}$ in dynamical regimes between 20 hPa·day$^{-1}$ and -100 hPa·day$^{-1}$: standard deviation is
around 2.5 % for $C_{Opaque}^{\boxplus}$, less than 2 % for $C_{Thin}^{\boxplus}$, around 2.5 K for $T_{Opaque}^{\boxplus}$, less than 3 K for $T_{Thin}^{\boxplus}$, approximately 0.01 for
$\varepsilon_{Thin}$, around 350 m for $Z_{Opaque}^{\boxplus}$ and $Z_{Base}^{\boxplus}$, 300 m for $Z_{Top}^{\boxplus} - Z_{Opaque}^{\boxplus}$ and 200 m for $Z_{Top}^{\boxplus} - Z_{Base}^{\boxplus}$. So, a change in the
large-scale dynamic regimes produces a change in the cloud properties and CRE that seem predictable. For example, if an
intensification of the upward air motions velocity change $\omega_{500}$ on a region from -40 hPa·day$^{-1}$ to -80 hPa·day$^{-1}$, $C_{Opaque}^{\boxplus}$
would increase by 8 % ($C_{Thin}^{\boxplus}$ will remain more or less constant), $T_{Opaque}^{\boxplus}$ will decrease by 10 K and $T_{Thin}^{\boxplus}$ by 7 K, and $\varepsilon_{Thin}$
will increase by 0.03. These cloud changes would increase the CRE by 17 W·m$^{-2}$ including 14 W·m$^{-2}$ from Opaque clouds
(Fig. 11f). Because $C_{Thin}^{\boxplus}$ will remain more or less constant whereas $C_{Opaque}^{\boxplus}$ will increase with a decrease of $\omega_{500}$ in
ascending regime, the relative contribution of Opaque clouds to the total CRE will be more and more important as
convection increases. This is why we see in Fig. 11g a decrease of the Thin clouds relative contribution from 20 hPa·day$^{-1}$ to
-100 hPa·day$^{-1}$.





FIG. 11. Tropical mean cloud properties and radiative effects as a function of the 500-hPa pressure velocity: (a) $C_{Opaque}^{\boxplus}$ and $C_{Thin}^{\boxplus}$, (b) $T_{Opaque}^{\boxplus}$ among Opaque clouds and $T_{Thin}^{\boxplus}$ among Thin clouds, (c) $Z_{Opaque}^{\boxplus}$ among Opaque clouds and $Z_{Base}^{\boxplus}$ among Thin clouds, (d) $Z_{Top}^{\boxplus} - Z_{Opaque}^{\boxplus}$ among Opaque clouds and $Z_{Top}^{\boxplus} - Z_{Base}^{\boxplus}$ among Thin clouds, (e) $\varepsilon_{Thin}^{\boxplus}$ among Thin clouds, (f) total CRE, Opaque CRE and Thin CRE and (g) relative contribution of Opaque CRE and Thin CRE. (h) Distribution of the 500-hPa pressure velocity. Results obtained from monthly mean 2°×2° gridded variables. Only nighttime over ice-free oceans for the 2008–2015 period in [30°S–30°N] is considered. The error bars show the ± standard deviation of the 96-monthly means.

Because cloud properties seem to be invariants for dynamical regimes, a change in the tropics of the large-scale circulation should provide a change in the CRE predictable and linked to the spatial distribution (both covers and altitudes) of Opaque clouds and Thin clouds sounded by CALIOP. For example, under global warming, climate models suggest a



narrowing of the ascending branch of the Hadley cell (e.g. Su et al., 2014), which means less convective regions and more

subsiding regions and which should result in a decrease of the CRE predictable knowing the changes of $\omega_{500}$ all over the

tropics.



## 6 Limitations of the OLR linear expression

In this study, from the direct measurement of the atmosphere opacity by spaceborne lidar, termed $Z_{Opaque}^{|}$, we were able to infer the radiative temperature of Opaque clouds $T_{Opaque}^{|}$, which we found linearly linked to the OLR. We propose $Z_{Opaque}^{|}$ as a good candidate to provide an observational constraint on the LW CRE. We tested the linear relationship at different space scales from instantaneous to monthly means. Hereinbelow, we list possible reasons for uncertainties.

### 6.1 Cloud radiative temperatures $T_{Opaque}^{|}$ and $T_{Thin}^{|}$

Cloud radiative temperatures $T_{Opaque}^{|}$ and $T_{Thin}^{|}$ definitions (Section 3.1) only take into account the apparent cloud extremities seen by the lidar ($Z_{Top}^{|}$ and $Z_{Opaque}^{|}$ or $Z_{Base}^{|}$). A temperature defined by the centroid altitude (Garnier et al., 2012) would take into account the entire cloud vertical profile. It could estimate better the equivalent radiative temperature. However, our results show that the CRE is mainly driven by $Z_{Opaque}^{|}$ and $Z_{Top}^{|}$ over Opaque clouds and $Z_{Base}^{|}$ and $Z_{Top}^{|}$ over Thin clouds. Furthermore, observational-based studies from the Atmospheric InfraRed Sounder (AIRS) and CALIOP showed that the radiative cloud height is located at the "apparent middle" of the cloud (Stubenrauch et al., 2010). The authors defining the "apparent middle" of the cloud as the middle between the cloud top ($Z_{Top}^{|}$) and the "apparent" cloud base sees by the CALIOP lidar ($Z_{Base}^{|}$ for Thin clouds and $Z_{Opaque}^{|}$ for Opaque clouds), consistently with our own definitions (Eqs. (1) and (2)).

### 6.2 Evaluation of the OLR over Thin clouds

We saw that the theoretical linear expression of $OLR_{Thin}^{|}$ for a fixed $\varepsilon_{Thin}^{|}$ overestimates the simulated one, up to $+10$ W·m$^{-2}$ for many cases (Section 4.1). This is partly due to the fact that the linear theoretical expression does not take into account the diffusion of the LW radiation within the clouds. It could partly explain why $OLR_{Thin}^{\oslash\,(LID)}$ is large compared to the measured $OLR_{Thin}^{\oslash\,(CERES)}$ (Fig. 6b). However, we do not think it should really affect the global scale partition of $CRE_{Total}^{\boxplus\,(LID)}$ between $CRE_{Opaque}^{\boxplus\,(LID)}$ and $CRE_{Thin}^{\boxplus\,(LID)}$, because, replacing $CRE_{Thin}^{\boxplus\,(LID)}$ by the difference $CRE_{Total}^{\boxplus\,(CERES)} - CRE_{Opaque}^{\boxplus\,(LID)}$, reveals that Opaque clouds contribute to 74 % to the total CRE instead of 73 %.

Plotting results of Fig. 6 in single-cloud-layer situations (not shown) shows better correlation coefficients, with R = 0.99 for Opaque clouds and R = 0.92 for Thin clouds. It reveals that our linear expression can be affected by additional uncertainties in multilayers situations. As an example, all the occurrences far from and over the identity line in Fig. 6a are due to cloud multilayers. For Opaque cloud single columns, taking into account the optical depth of the thinner cloud which overlaps an Opaque cloud in the expression of $T_{Opaque}^{|}$ improves the results (R = 0.97). However, this subtlety adds complexity to compute $T_{Opaque}^{|}$, and only gives small improvements to a simple expression with already very satisfying results (R = 0.95 on Fig. 6a).

Also, the value of $\varepsilon_{Thin}^{\boxplus}$ used to construct $OLR_{Thin}^{\boxplus\,(LID)}$ does not account for Thin cloud single columns where no "Clear" bin is found below the cloud (these clouds are not present in the $\varepsilon_{Thin}^{|}$ PDFs of Fig. 4d). This happens when very low clouds are present in the lowest 480 m bin. So, emissivities of Thin clouds close to the surface are not taken into account in the averaged $\varepsilon_{Thin}^{\boxplus}$. But since all these "missed" cloud emissivities are from clouds near the surface, their temperature is certainly close to the surface temperature and their LW CRE should be small. So, this effect should have no significant impact on the presented results.




Moreover, applying $OLR^|_{Thin}$ Eq. (4) to 2°×2° gridded variables introduces errors since the equation is non-linear
(product of $T^|_{Thin}$ and $\varepsilon^|_{Thin}$) unlike the $OLR^|_{Opaque}$ Eq. (5) which is linearly dependent on $T^|_{Opaque}$. Given that $\varepsilon^|_{Thin}$ is mostly
centered around 0.25 (Fig. 4d) it should not bring a substantial error, and the comparison of the computed gridded
$OLR^{⊞\,(LID)}_{Total}$ against measured $OLR^{⊞\,(CERES)}_{Total}$ has shown very good agreement.
Finally, due to the fact that the GOCCP product was built in order to avoid false cloud detections, the threshold
chosen for cloud detection implies that GOCCP does not detect high clouds with an optical depth smaller than about 0.07
(Chepfer et al., 2010, 2013). These subvisible cirrus clouds are not included in this study, but as their emissivities are very
small (smaller than about 0.03), they will likely not change the results of the paper.
**6.3 Gridded OLR**
Concerning gridded OLR, it should be noted that we used monthly mean $OLR^{⊞}_{Clear}$ from CERES-EBAF in Eqs. (4-
5) instead of instantaneous $OLR^{⊞}_{Clear}$ from C3M since this product is only available up to April 2011. Clear sky OLR from
CERES-EBAF data is derived only from measurements over Clear sky atmospheric columns which are generally drier than
the clear part of a cloudy atmospheric column. Then, because a drier atmospheric column leads to a stronger OLR (e.g.
Spencer and Braswell, 1997; Dessler et al., 2008; Roca et al., 2012), $OLR^{⊞}_{Clear}$ from CERES-EBAF should overestimates
$OLR^{⊞}_{Clear}$ from C3M in average. The diurnal cycle, which is taken into account in $OLR^{⊞}_{Clear}$ from CERES-EBAF but not in
$OLR^{⊞}_{Clear}$ from C3M (since we only used nighttime observations) could also play a role in the difference. We found an
increase of 0.6 W·m⁻² for the global mean $OLR^{⊞\,(LID)}_{Total}$ computed with $OLR^{⊞}_{Clear}$ from CERES-EBAF compared to $OLR^{⊞\,(LID)}_{Total}$
computed with $OLR^{⊞}_{Clear}$ from C3M for the 2008–2010 period.
Differences could also be related, to multilayer clouds in atmospheric single columns, to microphysics cloud
properties, and to differences in local atmospheric properties. However, using this very simple expression of the OLR give
an excellent correlation (R = 0.95) between monthly mean $OLR^{⊞\,(LID)}_{Total}$ and $OLR^{⊞\,(CERES)}_{Total}$ and a good agreement of the linear
regression with the identity line (appendix C, 2D distribution of monthly means 2°×2° gridded measured and computed OLR
is given in Fig. A4).
**6.4 Sensitivity to $Z^|_{Opaque}$ and to the multiple scattering factor**
We also checked the sensitivity of $OLR^{⊞\,(LID)}_{Total}$ to the uncertainty in the altitude of full attenuation of the lidar. To do
this, we computed the $OLR^{⊞\,(LID)}_{Total}$ assuming $Z^|_{Opaque}$ in all Opaque single column is located one bin (480 m) higher than
$Z^|_{Opaque}$ given by GOCCP v3.0. This leads to a modification of the Opaque cloud radiative temperature and then to a
modification of the $OLR^{|\,(LID)}_{Opaque}$ and so $OLR^{⊞\,(LID)}_{Total}$. Doing this, decreases the global mean $OLR^{⊞\,(LID)}_{Total}$ from 0.9 W·m⁻²
(appendix D, Fig. A5a).
Finally, the use of a fixed multiple scattering factor $\eta$ for the retrieving of the Thin cloud emissivity, whereas it
depends on cloud temperature (Garnier et al., 2015), could also play an important role in the differences between computed
$OLR^{⊘\,(LID)}_{Thin}$ and measured $OLR^{⊘\,(CERES)}_{Thin}$. We tested the sensitivity of a change in $\eta$ on $OLR^{⊞\,(LID)}_{Total}$, modifying the value of $\eta$
from 0.6 to 0.5. It reduces the global mean $OLR^{⊞\,(LID)}_{Total}$ from 1.1 W·m⁻² (appendix D, Fig. A5b), which we consider
negligible compared to the global mean value of $CRE^{⊞\,(LID)}_{Total}$ equal to 28.4 W·m⁻².



### 7 Conclusion


Simple radiative transfer models that estimate the top of the atmosphere outgoing radiations as a function of a
limited number of variables are useful tools to build first-order decomposition of climate feedbacks. Such simple models
exist in the SW domain, but not in the LW domain because the LW fluxes are sensitive to the cloud vertical distribution
making the definition of such a simple model more challenging in the LW than in the SW. In this work, we propose a simple
LW radiative model which express the LW CRE as a function of five variables: two of them describe the Opaque clouds
(Opaque cloud cover, Opaque cloud radiative temperature) and three others describe the semi-transparent clouds (Thin cloud
cover, Thin cloud radiative temperature and Thin cloud emissivity).
The originality of the approach proposed in this paper relies on how the cloud vertical distribution is described in
this simple radiative transfer model. We used three levels of altitude documented by a space borne lidar to describe the cloud
vertical distribution within the simple radiative model. Our approach contrasts with the techniques based on passive space
borne sensors because those latter measure vertically integrated variables and do not provide direct information on the cloud
vertical distribution. Our approach also contrasts with techniques based on lidar/radar measurements that use 40 levels of
altitude (or more) to describe the cloud vertical distribution in the troposphere. In this work, we take advantage of the
precision and accuracy of the space borne lidar to describe the cloud vertical structure, but we retain only three levels of
altitude out of the 40 or more, to describe the cloud vertical distribution. Considering only three levels of altitude allows to
build simple radiative models useful for first-order cloud feedback analysis, given that the more complex radiative transfer
models using 40 altitude levels can hardly be used for this purpose. The three levels of altitude that we have selected are the
ones which influence the most the OLR: 1) the cloud top altitude $Z^|_{Top}$ 2) the level of full attenuation of the lidar laser beam
$Z^|_{Opaque}$ in a single column containing an Opaque cloud, and 3) the cloud base $Z^|_{Base}$ in a single column containing a semi-
transparent Thin cloud. These three levels of altitudes have two advantages: they are first order drivers of the LW CRE and
they have been measured precisely and unambiguously over a decade with the CALIPSO space-borne lidar.
Using radiative transfer computations, we found that the OLR above an opaque cloud can be expressed linearly as a
function of the Opaque temperature: $OLR^{|(LID)}_{Opaque} = 2.0T^|_{Opaque} - 310$, where $T^|_{Opaque}$ is obtained from the combination of
the cloud top altitude $Z^|_{Top}$, the level of full attenuation of the lidar laser beam $Z^|_{Opaque}$, and a temperature profile from
reanalysis. From this simple relationship, it results that if an Opaque cloud rises up, and so decreases its $T^|_{Opaque}$ by 1 K, then
the OLR is decreased by 2 W·m⁻². Using this linear relationship together with CALIPSO and CERES observations, we
estimated the contribution of the Opaque clouds to the global mean LW CRE. Opaque clouds, which cover 35 % of the ice-
free ocean, contribute to 73 % of the global mean cloud radiative effect whereas Thin clouds, which cover 36 %, contribute
to 27 %.
We checked the robustness of the linear relationship given here above against observations at two different space
and time scales. First, we tested the instantaneous time scale at small space scale (20 km) using CALIPSO lidar data
collocated with CERES broadband radiometer data. We found a correlation coefficient of 0.95 between the lidar derived
$T^{\oslash}_{Opaque}$ and the OLR measured by the broadband radiometer. Second, we tested the validity of the relationship using
monthly mean data within 2° latitude × 2° longitude grid boxes. There we found that the global annual mean OLR derived
from the combination of the lidar data and the linear relationship, differs by 0.1 W·m⁻² from the OLR measured by CERES.
To conclude, this paper proposes a simple approximate formulation of the complex problem of radiative transfer in
the LW domain that could be used to explore first-order LW cloud feedback in both observations and climate model
simulations. On the observational side, future work will consist in analyzing the inter-annual variability of the record
collected by space-borne lidars and broadband radiometers: CALIPSO/CERES in the A-train (10+ years) completed by
EarthCare (Illingworth et al., 2014) to be launch in the coming years. On the climate model simulation side, this framework



will be included in the Cloud Feedback Model Intercomparison Project (CFMIP) Observation Simulator Package (COSP;
Bodas-Salcedo et al., 2011) lidar simulator (Chepfer et al., 2008) and applied to climate model outputs in order to quantify
the role of each cloud property in the simulated cloud feedbacks.





**Appendix A: Radiative cloud temperature**
Schematically, if we consider an optically uniform cloud, i.e. the LW optical depth $\delta^{LW|}$ increases linearly through
the cloud, with a cloud total LW optical depth $\delta_{Cloud}^{LW|}$, we can compute the upward LW radiative flux emitted by the cloud at
the top of the cloud ($\delta^{LW|} = 0$). Neglecting the cloud particle reflectivity in the longwave domain, from the integral form of
the Schwarzschild's equation, we can express the upward zenithal spectral radiance $I_\upsilon^{|}$ emitted by the cloud at the top of the
cloud:
$I_{\upsilon_{Cloud}}^{|}\left(\delta^{LW|} = 0\right) = \int_0^{\delta_{Cloud}^{LW|}} B_\upsilon\left(T\left(\delta^{LW|}\right)\right) e^{-\delta^{LW|}} d\delta^{LW|}$     [W·m$^{-2}$·sr$^{-1}$·m$^{-1}$] (A1)
Considering a linear increase of the temperature with $\delta^{LW|}$ from the cloud top to the cloud base ($T\left(\delta^{LW|}\right) =$
$k_1\delta^{LW|} + k_2$) and integrating $I_{\upsilon_{Cloud}}^{|}$ throughout the whole LW spectrum (using Stefan-Boltzmann law $\int B_\upsilon\, d\upsilon = \sigma T^4/\pi$),
we can write the LW radiance $I^{LW|}$ emitted by the cloud at the top of the cloud as:
$I_{Cloud}^{LW|}(\delta^{LW|} = 0) = \int_0^{\delta_{Cloud}^{LW|}} \frac{\sigma}{\pi}\left(k_1\delta^{LW|} + k_2\right)^4 e^{-\delta^{LW|}} d\delta^{LW|}$     [W·m$^{-2}$·sr$^{-1}$] (A2)
Assuming that the cloud emits as a Lambertian surface, the upward LW radiative flux $F^{\uparrow LW|}$ emitted by the cloud at
the top of the cloud is given by:
$F_{Cloud}^{\uparrow LW|}(\delta^{LW|} = 0) = \int_0^{\delta_{Cloud}^{LW|}} \sigma\left(k_1\delta^{LW|} + k_2\right)^4 e^{-\delta^{LW|}} d\delta^{LW|}$     [W·m$^{-2}$] (A3)
Then, for specific values of coefficient $k_1$ and $k_2$, which determine the gradient of temperature in the cloud and the
cloud top temperature (and so the cloud base temperature knowing $\delta_{Cloud}^{LW|}$), it is possible to compute $F_{Cloud}^{\uparrow LW|}\left(\delta^{LW|} = 0\right)$ and
then solve the equation $F_{Cloud}^{\uparrow LW|}(\delta^{LW|} = 0) = \varepsilon^{|}\sigma\left(T_{rad}^{|}\right)^4 = \left(1 - e^{-\delta_{Cloud}^{LW|}}\right)\sigma\left(T_{rad}^{|}\right)^4$ to find the corresponding equivalent
cloud radiative temperature $T_{rad}^{|}$.
**Appendix B: Vertical distributions of clouds directly observed by CALIOP**
For 3 regions, as for Fig. 4, Fig. A1 shows distributions of the distance between cloud top and $Z_{Opaque}^{|}$ among
Opaque clouds and the distance between cloud top and cloud base among Thin clouds. In the 3 regions, when an Opaque
cloud (Fig. A1a) is penetrated by the laser beam of the lidar, $Z_{Opaque}^{|}$ is mostly found in the 1$^{st}$ km below $Z_{Top}^{|}$ (30 % in the
tropical convective region, 52 % in the mid-latitudes region and 75 % in the tropical subsiding region). The frequency
distribution collapses after 1 km (note the logarithmic y-axis). The greater altitude differences between $Z_{Top}^{|}$ and $Z_{Opaque}^{|}$ can
be due to a more vertically spread cloud or to multiple cloud layers. If we look at the dashed lines, which represent the part
of the PDF considering only profiles without multilayers, we can see that the curves of the 3 regions fall to zero around 4–
5 km. This means that all the part of PDFs over 5 km are due to cloud multilayers. It also suggests that the laser beam never
sounds deeper than 5 km within a cloud.
Regarding Thin clouds (Fig. A1b), we mostly found $Z_{Base}^{|}$ in the 1$^{st}$ km below $Z_{Top}^{|}$ (49 % in the tropical convective
region, 68 % in the mid-latitudes region and 76 % in tropical subsiding region). The frequency distribution collapses after
1 km (again, note the logarithmic y-axis). The part of the PDF of profiles without multilayer (dashed lines), i.e. single
columns which contain only one optically thin cloud layer and so directly represent the geometrical thickness of Thin clouds,
fall to zero around 4–5 km. This means, as for Opaque clouds, that all the part of PDFs over 5 km are due to overlap of
multiple cloud layers. It therefore suggests, if we look at both Figs. A1a and A1b, that the laser beam is not able go through
the entire cloud if its vertical geometrical thickness is greater than 5 km. In other words, a cloud with a vertical geometrical
thickness greater than 5 km is always declared as an Opaque cloud. Furthermore, as PDFs collapse after 1 km in both figures



and for all regions, it also suggests that, even if the maximum penetration depth is 5 km, the laser beam is almost every time
totally attenuated when exceeding 1 km thickness.
**Appendix C: Verification of the lidar-derived gridded OLR against CERES observations**
Figure A4 shows the correlation between the OLR computed from lidar observations ($OLR_{Total}^{\boxplus \, (LID)}$) and the OLR
measured by the CERES radiometer on-board the Aqua satellite on which we extract only footprints collocated with the
CALIPSO ground track ($OLR_{Total}^{\boxplus \, (CERES)}$) for nighttime and over ice-free oceans on 2°×2° monthly means for the 2008. We
found an excellent correlation (R = 0.95) and the regression slope is near the one-to-one line which reinforces our confidence
in this simple OLR expression to correctly estimate the observed OLR.
**Appendix D: Sensitivity of the lidar-derived gridded OLR to $Z_{Opaque}^{|}$ and to the multiple scattering factor**
Figure A5a shows the difference between lidar-derived gridded $OLR_{Total}^{\boxplus \, (LID)}$ shown in Fig. 7a and the one which
would be obtain if $Z_{Opaque}^{|}$ was found 480 m higher. To do this, we replaced the altitude $Z_{Opaque}^{|}$ of each Opaque cloud
single column found with the lidar by the bin above, so the altitude of $Z_{Opaque}^{|}$ is systematically increased by 480 m. We
then recomputed $OLR_{Total}^{\boxplus \, (LID)}$ in the exact same way as described in this paper. The effect of an increase in the altitude of
$Z_{Opaque}^{|}$ is a global mean decrease in $OLR_{Total}^{\boxplus \, (LID)}$ by 0.9 W·m$^{-2}$. Areas where $OLR_{Total}^{\boxplus \, (LID)}$ is the most affected correspond to
areas with large values of Opaque cloud cover (patterns for 2008–2015 period on Fig. 2a are quite similar to those for the
year 2008) except for the stratocumulus regions off the West coasts of the African, the American and the Oceanian
continents where $C_{Opaque}^{\boxplus}$ is large but where $OLR_{Total}^{\boxplus \, (LID)}$ change is not very pronounced. A higher $Z_{Opaque}^{|}$ increases the level
of the radiative temperature of the Opaque clouds, so decreases this temperature and then weakens $OLR_{Total}^{\boxplus \, (LID)}$. Since
$OLR_{Total}^{\boxplus \, (LID)}$ is not affected as much in the stratocumulus regions, this suggests that vertical temperature gradient where these
clouds are founded must be weak.
Figure A5b shows the difference between lidar-derived gridded $OLR_{Total}^{\boxplus \, (LID)}$ shown in Fig. 7a and the one which is
obtain using a fixed multiple scattering factor $\eta = 0.5$ instead of $\eta = 0.6$. Decreasing $\eta$, increases the retrieved emissivity of
the Thin clouds by 0.05. Consequently, areas where Thin cloud cover is large and where they are high and cold, so where
they have a strong cloud radiative effect, are regions where $OLR_{Total}^{\boxplus \, (LID)}$ is the most affected by this change (in the multiple
scattering factor), up to a decrease of 3.5 W·m$^{-2}$ in the Indonesian region.



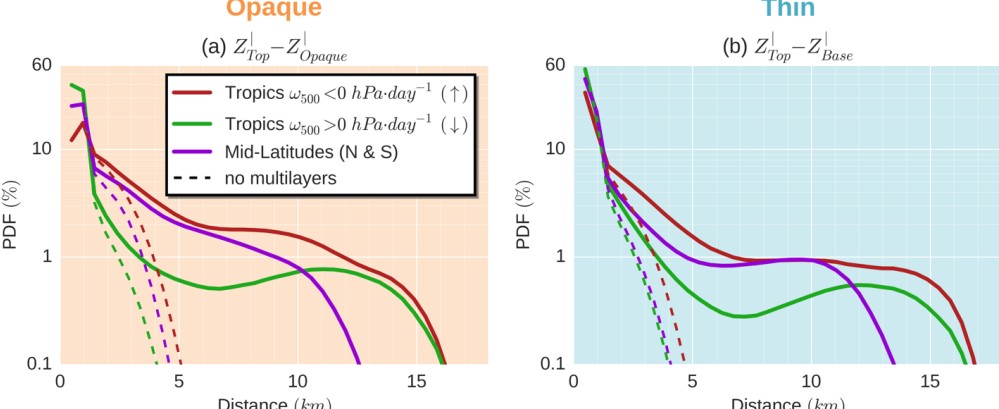

FIG. A1. Distributions of (a) the distance between cloud top and $Z^|_{Opaque}$ among Opaque clouds and (b) the distance between cloud top and cloud base among Thin clouds in three regions: same as Fig. 4. Dashed lines represent the distribution only among single columns where a unique cloud layer was found (no multiple cloud layers). Only nighttime over ice-free oceans for the 2008–2015 period is considered.

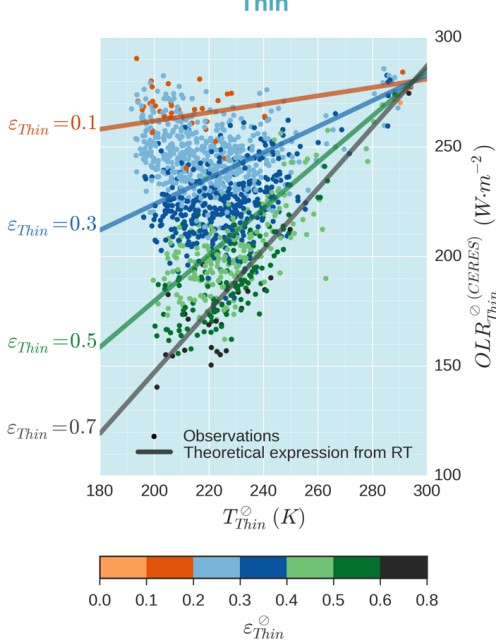

FIG. A2. Comparison between observed and lidar-derived OLR, at CERES footprint scale, as a function of $T^{\oslash}_{Thin}$ and $\varepsilon^{\oslash}_{Thin}$. Results obtained from CERES (dots) and CALIOP (lines) collocated measurements. Theoretical expressions are from Eq. (4). Same results as in Fig. 6b but only for measurements where $OLR^{\oslash}_{Clear}$ is close to 280 W·m⁻² selected ($OLR^{\oslash}_{Clear} \in [275-285]$ W·m⁻²), in order to only see the contribution of $T^{\oslash}_{Thin}$ and $\varepsilon^{\oslash}_{Thin}$ on the OLR. Only nighttime over ice-free oceans for January 2008 is considered.





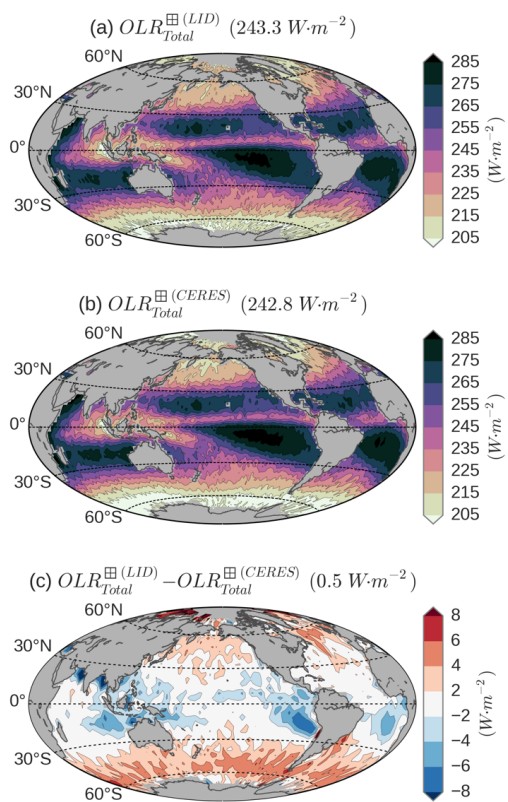


FIG. A3. Same as Fig. 7 but using $OLR_{Clear}^{\boxplus}$ from CERES-EBAF instead of $OLR_{Clear}^{\boxplus}$ from CERES-Aqua in the calculation of $OLR_{Total}^{\boxplus \, (LID)}$.


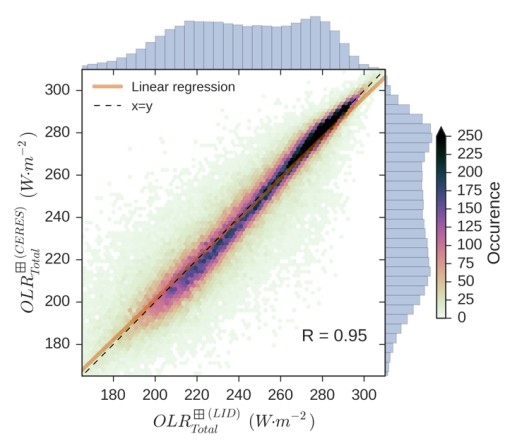


FIG. A4. Comparison between observed and lidar-derived OLR at monthly mean 2°×2° gridded scale. Only nighttime over ice-free oceans for the 2008-year period is considered.





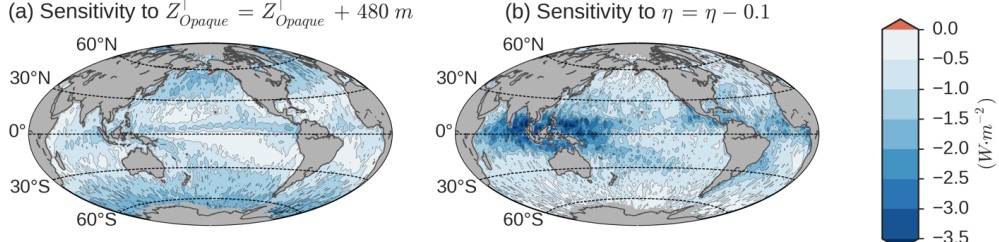

682  Fig. A5. Sensitivity of the lidar-derived annual-mean gridded $OLR_{Total}^{\boxplus\,(LID)}$ to the altitude of full attenuation of the lidar into Opaque

683  clouds $Z_{Opaque}^{|}$ and to the multiple scattering factor $\eta$: (a) difference between $OLR_{Total}^{\boxplus\,(LID)}$ of Fig. 7a and $OLR_{Total}^{\boxplus\,(LID)}$ which would be

684  obtain if $Z_{Opaque}^{|}$ was found a 480 m-bin upper and (b) difference between $OLR_{Total}^{\boxplus\,(LID)}$ of Fig. 7a and $OLR_{Total}^{\boxplus\,(LID)}$ which is obtain using a

685  fixed multiple scattering factor $\eta = 0.5$ instead of $\eta = 0.6$. Only nighttime over ice-free oceans for the 2008 year is considered.




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
