# Peer review of "Link between the Outgoing Longwave Radiation and the"

_Atmospheric Measurement Techniques, 2017_

## Referee Comment (RC1) · Anonymous Referee #2 · 4 Jul 2017

In this paper the authors devise a technique for relating – with a fairly high amount of accuracy – outgoing long wave radiation (OLR) at the top of the atmosphere (TOA) to several quantities that can be acquired from space-borne lidar (i.e., CALIOP on board Calipso). These quantities are the the radiative temperature and spatial coverage of opaque clouds and the radiative temperature, spatial coverage, and LW emissivity of thin clouds. Opaque clouds are defined as those for which the lidar beam becomes fully attenuated within the cloud, and typically have LW optical depths exceeding 1.5-2.5. Thin clouds, with LW optical depths less than this threshold, are semi-transparent and do not fully attenuate the lidar beam. The authors derive a simple semi-empirical relationship in which OLR increases by 2 W/m2 for every 1 K increase in opaque cloud

radiating temperature. For thin clouds, this 2:1 relationship is scaled by the cloud LW emissivity. OLR inferred from the lidar-derived quantities compares well with that measured directly by CERES, at a variety of spatial scales.

I found the technique described in the paper to be a clever use of the unique measurements provided by active sensors in space. Despite the presence of errors (notably for thin clouds), the OLR can be largely reproduced from 5 basic measurements, which makes it a powerful tool for relating cloud property changes to OLR. I recommend publication pending revisions based on the my concerns that are detailed below.

Major Comments: 1) My main concern with this work is that the authors may be slightly overstating the value of such an analysis, especially in regard to how it is contrasted with passive sensors. Passive sensors are rightfully criticized for often giving incorrect information about cloud vertical distribution, which active sensors retrieve with much higher accuracy. However, passive sensors are (essentially) directly retrieving the quantity that the authors need to derive here: the emission temperature of clouds. Passive retrievals may not place the cloud top at the correct physical altitude like a lidar does, but they do place it at the effective radiating temperature, which is what matters for the OLR and any TOA LW anomalies. This is basically what makes studies that relate TOA radiation to passive-derived cloud fraction histograms like Hartmann et al. (1992), Zelinka et al DOI: 10.1175/JCLI-D-11-00248.1 (2012) and Yue et al DOI: 10.1175/JCLI-D-15-0257.1, (2016) possible. The authors are sort of reverse-engineering this problem: They have highly accurate measurements of backscatter by cloud particles as a function of altitude, which they then use in a clever way to derive the effective radiating temperature, which is what you would already have if you started with passive measurements. It is not obvious to me that this is superior. I think the paper requires a clear discussion of why one would prefer this technique over one relying directly on passive measurements, and/or a discussion of how they both could complement each other. Simply asserting that active sensors retrieve the vertical profile of condensate more accurately is not compelling in this particular context. One advantage I can think of relative to existing kernel techniques is that it does indeed seem desirable to have a small set of measurements that one can get both from observations (Calipso) and models (albeit, those running the Calipso simulator) that can give a highly accurate proxy for OLR, in keeping with the analogy to APRP in the SW. This is in contrast to relying on 7x7 histogram of cloud types from ISCCP and a kernel to match. Perhaps another advantage has to do with the more practical issue of observing cloud changes over a long period of time. Few people trust ISCCP trends because of various issues that arise with splicing many individual satellites together that are poorly inter-calibrated and have non-climate related trends from satellite orbit changes, view angle changes, etc. (Norris and Evan DOI: 10.1175/JTECH-D-14-00058.1 2015). Presumably some of these issues are less relevant for lidars? If so, it would be important to distinguish these sorts of problems from those arising from the retrieval philosophy (e.g., if ISCCP was a perfect system without any artifacts, would the active approach still be superior?)

2) On lines 362-365, the authors state "Monitoring T_Opaque on longterm should provide important information which should help to better understand the LW cloud feedback mechanism. Moreover, because the relationship is linear, it simplifies the derivatives in mathematical expressions of feedback and will allow to construct a useful framework to study LW cloud feedback in simulations of climate models." Feedbacks are conventionally defined as the change in a given quantity holding all else fixed. In the case of altitude feedback, this would be the change in cloud altitude only, with everything including the temperature profile fixed. Mathematically, this is equivalent to comparing a control OLR with a hypothetical one computed with the cloud at a higher altitude and therefore at a lower emission temperature. Of course we know that in reality the cloud top temperature is expected to stay nearly constant with surface warming as the cloud top altitude rises with the isotherms (i.e., FAT hypothesis of Hartmann and Larson 2002). Changes in T_Opaque will depend on both the change in cloud altitude and the change in temperature profile, and constant T_Opaque may mean perfectly complementary changes in both the altitude and the temperature profile, as

one expects from FAT. If one uses your relationship between OLR and T_Opaque in computing feedbacks, then the mathematical formulation of the feedbacks will need to be changed to accommodate this. Specifically, I think one would need to compare the fixed T_Opqaue (FAT) case against a hypothetical baseline situation in which all things change except for the Z_Opaque, such that T_Opaque warms as much as a fixed altitude. While this is do-able, I disagree with the statement above that this simplifies the mathematics of feedbacks.

3) The English is very poor throughout the manuscript. There were far too many errors for me to list all of them (grammar, spelling, awkward phrasings, words that are plural that should not be, incorrect comma usage, etc.). In some places the writing was poor enough that the meaning of the sentence was unclear. This paper should be copyedited by a native English speaker before the reviewers see it again. In contrast, the figures were very clear, well-designed, and well-executed.

Minor Comments: In addition to the numerous English errors, I note the following:

Title: I would suggest deleting "the" before Outgoing and also rephrasing to "...where a space borne-lidar..."

Throughout: "cloud altitude longwave" seems awkward. Please rephrase to "longwave cloud altitude"

Abstract: This ends very abruptly. It needs a better closing sentence.

Lines 29-34: An uninformed reader of this paragraph will assume that the only reason there is uncertainty in how clouds will respond to warming is because models simulate biased clouds in the mean state. Surely this is not the only reason for low confidence in cloud feedbacks. There are a variety of recent review articles out on cloud feedbacks that may be helpful on this point.

Lines 52-54: This statement needs to be rephrased. Emergent constraints are not feedback mechanisms.

Lines 64-65: I disagree that there is no link between observed cloud variables and LW CRE. See, for example, the section on LW cloud altitude feedback in Ceppi et al doi: 10.1002/wcc.465 (2017), which points out that high cloud amount and emissivity, along with the temperature structure of the upper troposphere, govern the strength of this feedback. All of these are observable.

Lines 85-87: Cloud fraction histograms from passive sensors generally report cloud fraction on 7 cloud top pressure bins; the high, mid, and low aggregating is usually done later to simplify.

Lines 88-89: Suggest also citing Zhou et al DOI: 10.1175/JCLI-D-12-00547.1 (2013) and Yue et al 10.1002/2016JD025174 (2017), who have done this globally

Lines 90-91: These studies should be more clearly distinguished from the ones preceding it in the sentence: they have focused on trends, not interannual variability.

Line 97: Mace et al (2011) DOI: 10.1175/2010JCLI3517.1 should be cited here

Lines 168-170: I can't understand this. Please rephrase.

Line 183: should be "sea ice"

Line 185: Should be "Flux observations collocated with lidar cloud observations"

Line 216: Should "as" be "that"?

Figure 4: Is it possible to compare these cloud emission temperatures with those from passive sensors? They should be in agreement, right?

Line 273: "$T_{opaque}$ among opaque clouds" is redundant. This sort of statement occurs throughout the document.

Line 282: meaning of "mid-effect" is unclear

Line 288: "pick" should be "peak"

Line 303: rephrase

Lines 422-423: Rephrase.

Figure 8: Is the shading 2-sigma? Max to min?

Line 433: "under the tropics" – rephrase

Line 453: I don't know what this statement means.

Lines 488-493: The authors seem to be implying that omega is the only variable on which the cloud properties and CRE depend, and that therefore knowing how omega change will tell one how cloud properties and CRE will change. This is incorrect, as has been discussed many times over, most notably by Bony et al DOI 10.1007/s00382-003-0369-6 (2004) where this type of analysis originally appeared. While omega changes may strongly determine regional changes in cloud properties, when averaged over the entire tropics, it is the thermodynamic sensitivity of cloud propertiesÂǎwithin omega bins that emerges as the dominant driver of cloud changes.

Section 6.1: It is unclear whether this is actually an error source. The authors raise the issue then immediately downplay it. Is it a source of error? Have you actually performed a sensitivity study to determine with these assumptions matter?

Section 6.4: the impacts of these assumptions are being assessed on the global mean OLR, but I wonder whether they also influence the slope of OLR on T_Opaque.

---

## Referee Comment (RC2) · Anonymous Referee #3 · 6 Jul 2017

General

The authors present a methodology to estimate the outgoing longwave radiation (OLR) at the global scale from cloud products derived with the help of long-term space-borne measurements with the lidar CALIOP onboard CALIPSO. The major information comes from the opacity altitude of the atmosphere, i.e. the altitude at which the laser beam is fully attenuated due to clouds, and the geometrical cloud top height, which together allow the estimation of the radiative temperature of the cloud. It is shown that the latter one is linearly related to the OLR. Non-opaque (thin) clouds are treated in terms of top and base heights together with their emissivity, which is estimated from the lidar

attenuated scattering ratio below the cloud under consideration of a constant multiple-scattering factor. For opaque clouds, a very good correlation between the derived OLR and the one measured by CERES is found, whereas a systematic deviation is seen for thin clouds. Despite some possible explanations the reason for the deviation in the case of thin clouds does not finally become clear.

In general, the paper presents an interesting approach to study longwave radiation effects of clouds at the global scale. The paper deserves publication, but has the potential to be improved both in terms of scientific contents as well as style of presentation. I recommend publication after consideration of the comments below.

Major

My major concerns are related to the rather simplified approach of using only two cloud scenarios, namely single-layer thin and opaque clouds. I would at least expect an extended sensitivity study regarding more realistic scenes in the very beginning. Justifying the approach before the presentation and discussion of results would be much more satisfying for the reader than the currently provided discussion of limitations in Sec. 6 (where several questions are tackled which the reader has already in mind when reading the major part of the paper). In particular, the following cases need to be considered in the evaluation and discussion of obtained results throughout the paper, starting already in Sec. 2.1 and Fig. 1.

1) Multi-layer clouds: The discussion related to multi-layer clouds is not sufficient. The authors have added a very short paragraph in Sec. 2.1 (lines 176-179) during the technical revision of the paper. However, this explanation deals with thin clouds only. The more common feature is the appearance of thin, high cirrus clouds over mid-level or low-level opaque clouds. It is well known that retrievals from passive sensors locate the radiative cloud top height (or radiative temperature) in between the cloud layers in such cases, and that the location will depend on the optical thickness of the upper "thin" cloud. This fact is obviously not covered by the presented approach, since it

considers only the geometrical properties of cloud top height and opacity altitude for the calculation of the radiative temperature. Although some discussion is provided in Sec. 6.2, no substantial investigation of the related consequences for the approach is given.

2) Broken clouds: The authors find a high amount of "thin clouds" in the lower troposphere at temperatures above 0 °C, i.e. liquid clouds (Fig. 4). Usually, liquid clouds are not penetrated by lidar, even if they are geometrically thin (thickness of a few hundred meters). Those occasions of "thin clouds" might often be related to broken opaque clouds partly hit and partly missed by the lidar beam, thus leading to signals from the cloud and from the atmosphere and surface below the cloud in the same profile, so that the cloud appears to be transparent. The effect may be due to broken clouds within a single laser footprint, but can also result from averaging of laser shots over cloudy and clear atmospheric volumes before further retrievals are applied. From the description in Sec. 2.1, it does not become clear how averaging of lidar profiles is done, what exactly is meant with "each atmospheric single column" (line 127), which basic products (single shot, 1-km averages, 5-km averages) are used, and how the averaging to the 2°x2° grid is performed. It should be studied which differences in the results are expected when sub-scale broken opaque clouds instead of thin clouds appear. It would be interesting to see whether the worse correlation between calculated and measured OLR found for thin clouds could be explained in this way. In this context, also the discussion in Sec. 6.2 is insufficient.

Minor

Abstract: The abstract doesn't say anything about the retrievals for thin clouds.

Line 185, should be: "Flux observations collocated with lidar cloud observations"

Line 290, regarding the "second mode": What does "more diffuse" mean? What about altocumulus, altostratus clouds?

[Figure]

Line 300, "cloud emissivity of the cloud": correct to either "cloud emissivity" or "emissivity of the cloud".

Lines 331-332, "in spite of significant differences in the atmospheric temperature and humidity profiles": What does "significant" mean? How are these differences considered/validated in the calculations?

Line 372, "The evaluation . . . is only using observation from January 2008": This explanation should be given in the beginning of the discussion of Fig. 6.

Lines 405-415: Explain the units to be applied in the equations.

Lines 556 and 561, "decreases. . .from. . .", "reduces. . .from. . .": The meaning of the sentences with the word "from" is unclear.

There a many language/grammar/punctuation errors, which cannot be listed in detail here. The manuscript needs careful copy editing.

---

## Referee Comment (RC3) · Anonymous Referee #1 · 1 Aug 2017

**Review of de Guélis et al.: Link between the Outgoing Longwave Radiation and the altitude where the space-borne lidar beam is fully attenuated**

In this paper, which appears to be a follow-on from Guzman et al., 2017, the authors develop a simple approximation that allows them to estimate outgoing longwave radiation (OLR) using three parameters that are readily obtained from space-based lidar measurements: cloud top, cloud base (or, for opaque layers, apparent base) and cloud optical depths. Cloud altitudes are converted to temperatures using model data. The optical depths are used to compute emissivities. Since the current generation of space-based lidars cannot measure the optical depth of opaque layers, the emissivities for these clouds are assumed to be 1. For opaque clouds, OLR is approximated as a simple linear function of mid-layer temperature. The approximation for transparent clouds also uses mid-layer temperature, but is not as straightforward, as it also requires estimates of cloud emissivity and the OLR in clear sky conditions. Collocated CERES measurements are used to characterize the accuracy of both approximations.

The material presented in this paper is appropriate for AMT, and, after a few modifications are made, I believe the manuscript should eventually be published. The English language usage is, at times, somewhat (and occasionally very) awkward; however, the paper is well-organized, the figures are well-done and informative, the authors' derivation of their technique was clear and the steps taken to verify its performance were appropriate and straightforward. While the most interesting (and potentially useful) part of the manuscript was section 6, where the authors describe the limitations of their method, there are still a couple of issues that I believe deserve further investigation.

1. I had hoped to find a clear and convincing explanation for the rotation of the thin cloud data from the one-to-one line that is so evident in Figure 6b. In particular,

    (a) I'd like to know if this rotation is diminished in the "single-cloud-layer situations (not shown)", for which R increases from 0.89 to 0.92 (I suggest including the "not shown" plots in a future revision);

    (b) I'm intrigued by the differences in the sampling distributions for the opaque clouds vs. the thin clouds. For opaque layers, there is a noticeable skew in the distribution caused by (per line 518) "occurrences far from and over the identity line in Fig. 6a". But for the thin clouds in Fig. 6b the sampling distribution appears to be normally distributed about a single straight line). Do the authors have any thoughts or speculations about the root cause(s) for this difference in behavior?

2. How sensitive is the thin cloud OLR to emissivity errors introduced by aerosol contamination of "clear air" beneath the clouds detected by GOCCP?

Minor issues:

Line 17 : how much does the "atmosphere opacity altitude" depend on the (a) capabilities of the lidar used to measure the cloud, (b) the ambient lighting conditions, and (c) the algorithms used to retrieve apparent cloud base?

Lines 126–175 : nothing in this description makes it clear that columns containing multiple layers are actually included in the analyses. The fact that all columns are partitioned into one of the

three categories (i.e., clear, thin cloud, and opaque cloud) should be made clear from the very beginning, and not postponed until lines 176–179.

Line 171 : in the vast majority of CALIPSO literature (including Garnier et al., 2015, which is cited here), the symbol for optical depth is $\tau$. $\delta$ is used for depolarization ratios.

Lines 378–383 : here and elsewhere, I find the authors' notation to be very complex and cumbersome, which makes the text difficult to read and hard to understand.

Lines 530–531 : to my eye, the midlatitude emissivities are not "mostly centered around 0.25"

Line 554 : according to my (admittedly limited) understanding of the way the GOCCP cloud detection scheme works, a more realistic assessment would have been obtained by using on bin lower rather than one bin higher.

Lines 641–642 : the suggestion that "the laser beam is not able go through the entire cloud if its vertical geometrical thickness is greater than 5 km" is demonstrably false. For example, see

https://www-calipso.larc.nasa.gov/products/lidar/browse_images/show_detail.php?s= production&v=V4-10&browse_date=2010-01-01&orbit_time=12-47-14&page=3& granule_name=CAL_LID_L1-Standard-V4-10.2010-01-01T12-47-14ZN.hdf

The region between ~1.6° S and ~5.4° S contains numerous examples of transparent cirrus that are more than 6 km thick.

---

## Author Comment (AC1) · 20 Sep 2017

The comment was uploaded in the form of a supplement:
https://www.atmos-meas-tech-discuss.net/amt-2017-115/amt-2017-115-AC1-
supplement.pdf

---

## Author Response (AR1)

**Point-by-point response to the reviews**
* * *
**Anonymous Referee #1**

In this paper, which appears to be a follow-on from Guzman et al., 2017, the authors develop a simple approximation that allows them to estimate outgoing longwave radiation (OLR) using three parameters that are readily obtained from space-based lidar measurements: cloud top, cloud base (or, for opaque layers, apparent base) and cloud optical depths. Cloud altitudes are converted to temperatures using model data. The optical depths are used to compute emissivities. Since the current generation of space-based lidars cannot measure the optical depth of opaque layers, the emissivities for these clouds are assumed to be 1. For opaque clouds, OLR is approximated as a simple linear function of mid-layer temperature. The approximation for transparent clouds also uses mid-layer temperature, but is not as straightforward, as it also requires estimates of cloud emissivity and the OLR in clear sky conditions. Collocated CERES measurements are used to characterize the accuracy of both approximations.

The material presented in this paper is appropriate for AMT, and, after a few modifications are made, I believe the manuscript should eventually be published. The English language usage is, at times, somewhat (and occasionally very) awkward; however, the paper is well-organized, the figures are well-done and informative, the authors' derivation of their technique was clear and the steps taken to verify its performance were appropriate and straightforward. While the most interesting (and potentially useful) part of the manuscript was section 6, where the authors describe the limitations of their method, there are still a couple of issues that I believe deserve further investigation.

1. I had hoped to find a clear and convincing explanation for the rotation of the thin cloud data from the one-to-one line that is so evident in Figure 6b.

- **Response:**
  - The rotation of the thin cloud data from the one-to-one line does not affect the results of this study. Indeed, we did a sensitivity study to $CRE_{Thin}^{\boxplus\,(LID)}$ (Sect. 6.3): instead of computing the lidar-derived $CRE_{Thin}^{\boxplus\,(LID)}$ using the relationship used in Fig. 6b, we consider $CRE_{Thin}^{\boxplus}$ as the residual between CERES-derived total $CRE_{Total}^{\boxplus\,(CERES)}$ and lidar-derived $CRE_{Opaque}^{\boxplus\,(LID)}$: $CRE_{Thin}^{\boxplus} = CRE_{Total}^{\boxplus\,(CERES)} - CRE_{Opaque}^{\boxplus\,(LID)}$. This leads to Opaque clouds contributing to 74 % to the total CRE instead of 73 % in global mean. It is then not sensitive.
  - The rotation of the thin cloud data from the one-to-one line is the consequence of multiple effects. We examine hereafter the points raised by the reviewer. Thank you.

  In particular,

  (a) I'd like to know if this rotation is diminished in the "single-cloud-layer situations (not shown)", for which R increases from 0.89 to 0.92 (I suggest including the "not shown" plots in a future revision);

- **Response:** This rotation is not diminished in the "single-cloud-layer situations" (Fig. A4d).
- **Change made:**
  - In Sect. 6: Sect. 6.2 Multi-layer cloud and broken cloud situations has been added.
  - In Appendix: Fig. A4 has been added. It shows the decomposition of Fig. 6 in "single-layer cloud" and "multi-layer cloud" situations. The main text refers to Fig. A4 in Sect. 6.2.

  (b) I'm intrigued by the differences in the sampling distributions for the opaque clouds vs. the thin clouds. For opaque layers, there is a noticeable skew in the distribution caused by (per line 518) "occurrences far from and over the identity line in Fig. 6a". But for the thin clouds in Fig. 6b the sampling distribution appears to be normally distributed about a single straight line). Do the authors have any thoughts or speculations about the root cause(s) for this difference in behavior?

- **Response:** The new Fig. A4e shows that the noticeable skew in the distribution is due to multi-layer cloud situations. In these situations, an optically thin cloud overlapping an optically opaque cloud will tend to significantly underestimate $T_{Opaque}^{|}$ as we do not consider the difference of emissivity between the two clouds. For thin clouds, in presence of multi-layer cloud situations (Fig. A4f), $T_{Thin}^{|}$ can be overestimated or underestimated depending on which cloud is optically thicker. The contrast between their emissivity is generally smaller than for an opaque multi-layer cloud situation. This is the reason why there is no noticeable skew in the distribution for the thin clouds.

2. How sensitive is the thin cloud OLR to emissivity errors introduced by aerosol contamination of "clear air" beneath the clouds detected by GOCCP?

- **Response:** The computation of the Thin cloud emissivity $\varepsilon_{Thin}^{|}$ used all the clear sky layers (without aerosol) located below the lowest cloud layer, in order to determine the optical thickness of the cloud layers. If, for example, an aerosol layer is present just below the cloud, $\varepsilon_{Thin}^{|}$ would be derived from the sum of the cloud layer optical thickness and the aerosol layer optical thickness. As this study is only over ocean, errors introduced by aerosol are essentially found during boreal summer over a limited area: the dust plume (Peyridieu et al., 2010 DOI:10.5194/acp-10-1953-2010). Moreover, with regards to this study, we are interested in CRE, which, over ocean, are far larger than aerosol direct radiative effect.

**Minor issues:**

Line 17 : how much does the "atmosphere opacity altitude" depend on the (a) capabilities of the lidar used to measure the cloud, (b) the ambient lighting conditions, and (c) the algorithms used to retrieve apparent cloud base?

- **Response:** The "atmosphere opacity altitude" $Z_{Opaque}^{|}$ indeed depends on these three aspect.
    - (a) The accuracy of $Z_{Opaque}^{|}$ depends on the vertical resolution of the lidar, the telescope field of view, and the capabilities receiver sensor (noise). These uncertainty sources likely give error smaller than one 480 m bin.
    - (b) $Z_{Opaque}^{|}$ retrieval is difficult during daytime because daytime conditions are much noisier than the nighttime conditions in CALIOP data. This is the reason why we only use nighttime data in this study.
    - (c) $Z_{Opaque}^{|}$ depends on the algorithm used to retrieve apparent cloud base. It depends on the horizontal and vertical averaging choice (Chepfer et al., 2013; Cesana et al., 2016).
        - · Chepfer et al. (2013) – DOI:10.1175/JTECH-D-12-00057.1
        - · Cesana et al. (2016) – DOI:10.1002/2015JD024334
- **Change made:**
    - ➢ In Sect. 2.1 (1st §): "$Z_{Opaque}^{|}$ depends on the horizontal and vertical averaging used in the retrieval algorithm. It is also affected by sunlight noise during daytime. At 480 m vertical resolution, it poorly depends on the lidar characteristics." has been added.

Lines 126–175 : nothing in this description makes it clear that columns containing multiple layers are actually included in the analyses. The fact that all columns are partitioned into one of the three categories (i.e., clear, thin cloud, and opaque cloud) should be made clear from the very beginning, and not postponed until lines 176–179.

- **Change made:**
    - ➢ In Sect. 2.1 (1st §): "The GCM-Oriented CALIPSO Cloud Product (GOCCP)-OPAQ (GOCCP v3.0; Guzman et al., 2017) segregates each atmospheric single column sounded by the CALIOP lidar as one of the 3 following single column types" has been replaced by "The GCM-Oriented CALIPSO Cloud Product (GOCCP)-OPAQ (GOCCP v3.0; Guzman et al., 2017) has 40 vertical levels with 480 m vertical resolution. Every CALIOP single shot profile — including multi-layer profiles — is classified into one of three types".

Line 171 : in the vast majority of CALIPSO literature (including Garnier et al., 2015, which is cited here), the symbol for optical depth is τ. δ is used for depolarization ratios.

- **Response:** We agree with the reviewer.
- **Change made:**
    - ➢ Throughout the paper: "$\delta$" has been replaced by "$\tau$".

Lines 378–383 : here and elsewhere, I find the authors' notation to be very complex and cumbersome, which makes the text difficult to read and hard to understand.

- **Response:** We agree with the reviewer that our notation can be sometimes cumbersome. However, we choose this very explicit notation in order to avoid misleading interpretation as, throughout the paper, calculations are made at different spatial resolution (lidar single shot, CERES footprint, and gridded).

Lines 530–531 : to my eye, the midlatitude emissivities are not "mostly centered around 0.25"

- **Response:** We agree with the reviewer that this statement is not very accurate and has been removed.
- **Change made:**

> In Sect. 6.3 ($3^{rd}$ §): "Given that $\varepsilon_{Thin}^{|}$ is mostly centered around 0.25 (Fig. 4d) it should not bring a substantial error, and" has been replaced by "However,".

Line 554 : according to my (admittedly limited) understanding of the way the GOCCP cloud detection scheme works, a more realistic assessment would have been obtained by using on bin lower rather than one bin higher.

- **Response:** We choose to take one bin higher for the sensitivity test on $Z_{Opaque}^{|}$ in order to be able to apply this in the same way for every opaque cloud profile. Indeed, a non-negligible amount of opaque cloud profiles have their $Z_{Opaque}^{|}$ at the lowest GOCCP level (240 m above sea level), and taking the equivalent of a bin lower would have given negative opacity altitudes (–240 m). This problem is avoided taking one bin higher instead and the sensitivity test should not be sensitive to this choice since the relation between $OLR_{Opaque}^{|}$ and $T_{Opaque}^{|}$ is linear.
- **Change made:**
  > In Sect. 6.5 (first §): "(as moving $Z_{Opaque}^{|}$ one bin down would have led to negative values for some $Z_{Opaque}^{|}$)" has been added.

Lines 641–642 : the suggestion that "the laser beam is not able go through the entire cloud if its vertical geometrical thickness is greater than 5 km" is demonstrably false. For example, see
https://www-calipso.larc.nasa.gov/products/lidar/browse_images/show_detail.php?s=production&v=V4-10&browse_date=2010-01-01&orbit_time=12-47-14&page=3&granule_name=CAL_LID_L1-Standard-V4-10.2010-01-01T12-47-14ZN.hdf
The region between ~1.6° S and ~5.4° S contains numerous examples of transparent cirrus that are more than 6 km thick.

- **Response:** We agree with the reviewer.
- **Change made:**
  > In Appendix B ($2^{nd}$ §): "[…] the laser beam is not able go through the entire cloud if its vertical geometrical thickness is greater than 5 km […]" has been removed.

**Point-by-point response to the reviews**
* * *
**Anonymous Referee #2**

In this paper the authors devise a technique for relating – with a fairly high amount of accuracy – outgoing long wave radiation (OLR) at the top of the atmosphere (TOA) to several quantities that can be acquired from space-borne lidar (i.e., CALIOP on board Calipso). These quantities are the the radiative temperature and spatial coverage of opaque clouds and the radiative temperature, spatial coverage, and LW emissivity of thin clouds. Opaque clouds are defined as those for which the lidar beam becomes fully attenuated within the cloud, and typically have LW optical depths exceeding 1.5-2.5. Thin clouds, with LW optical depths less than this threshold, are semi-transparent and do not fully attenuate the lidar beam. The authors derive a simple semi-empirical relationship in which OLR increases by 2 W/m2 for every 1 K increase in opaque cloud radiating temperature. For thin clouds, this 2:1 relationship is scaled by the cloud LW emissivity. OLR inferred from the lidar-derived quantities compares well with that measured directly by CERES, at a variety of spatial scales.

I found the technique described in the paper to be a clever use of the unique measurements provided by active sensors in space. Despite the presence of errors (notably for thin clouds), the OLR can be largely reproduced from 5 basic measurements, which makes it a powerful tool for relating cloud property changes to OLR. I recommend publication pending revisions based on the my concerns that are detailed below.

**Major Comments:**

1) My main concern with this work is that the authors may be slightly overstating the value of such an analysis, especially in regard to how it is contrasted with passive sensors. Passive sensors are rightfully criticized for often giving incorrect information about cloud vertical distribution, which active sensors retrieve with much higher accuracy. However, passive sensors are (essentially) directly retrieving the quantity that the authors need to derive here: the emission temperature of clouds. Passive retrievals may not place the cloud top at the correct physical altitude like a lidar does, but they do place it at the effective radiating temperature, which is what matters for the OLR and any TOA LW anomalies. This is basically what makes studies that relate TOA radiation to passive-derived cloud fraction histograms like Hartmann et al. (1992), Zelinka et al DOI: 10.1175/JCLI-D-11-00248.1 (2012) and Yue et al DOI: 10.1175/JCLI-D-15-0257.1, (2016) possible. The authors are sort of reverse- engineering this problem: They have highly accurate measurements of backscatter by cloud particles as a function of altitude, which they then use in a clever way to derive the effective radiating temperature, which is what you would already have if you started with passive measurements. It is not obvious to me that this is superior. I think the paper requires a clear discussion of why one would prefer this technique over one relying directly on passive measurements, and/or a discussion of how they both could complement each other. Simply asserting that active sensors retrieve the vertical profile of condensate more accurately is not compelling in this particular context.

- ■ **Response:** We agree with the reviewer that a clear discussion of why one would prefer this technique over one relying on passive measurements is required. Thank you for your comment.
- ■ **Change made:**
  - ➤ In Sect. 3.1 (last §): "These cloud radiative temperatures are fundamental to study the LW CRE and are different from the effective radiating temperatures measured by passive instruments which are influenced by radiation coming from below the cloud. In the case of Opaque cloud which completely absorbs upward LW radiative flux propagating from below, the effective radiating temperature measured by passive instruments should agree with the cloud radiative temperature. However, this assumes to know that the cloud is Opaque, but cloud emissivity from passive measurements is also sensitive to hypothesis made on the clear sky and surface property. Unlike passive measurements, lidar measurements robustly separate Opaque clouds and Thin clouds from the presence or not of a surface echo (Guzman et al., 2017)." has been added.

One advantage I can think of relative to existing kernel techniques is that it does indeed seem desirable to have a small set of measurements that one can get both from observations (Calipso) and models (albeit, those running the Calipso simulator) that can give a highly accurate proxy for OLR, in keeping with the analogy to APRP in the SW. This is in contrast to relying on 7x7 histogram of cloud types from ISCCP and a kernel to match.

- ■ **Response:** Thank you for this comment.
- ■ **Change made:**
  - ➤ In Introduction (8[th] §): "We propose to build on these studies by adding the space-borne lidar information." has been replaced by "We propose to build on these studies by adding spaceborne lidar information to obtain a simplified radiative transfer model in the LW domain that can give a highly accurate proxy for OLR with a small set of parameters available from both observations (space-lidar) and models (space-lidar simulator). This approach is in contrast to reliance on 7×7 histograms (altitude×optical depth) of cloud types from ISCCP and use of a matching radiative kernel.".

Perhaps another advantage has to do with the more practical issue of observing cloud changes over a long period of time. Few people trust ISCCP trends because of various issues that arise with splicing many individual satellites together that are poorly inter-calibrated and have non-climate related trends from satellite orbit changes, view angle changes, etc. (Norris and Evan DOI: 10.1175/JTECH-D-14-00058.1 2015). Presumably some of these issues are less relevant for lidars? If so, it would be important to distinguish these sorts of problems from those arising from the retrieval philosophy (e.g., if ISCCP was a perfect system without any artifacts, would the active approach still be superior?)

- **Response:** Thank you for this comment.
- **Change made:**
  - In Introduction (8th §): "Moreover, a highly stable long-time observational record is essential to study clouds and climate feedback (Wielicki et al., 2013), and current passive instruments have shown limited calibration stability over decadal time scales (e.g. Evan et al., 2007; Norris and Evan, 2015; Shea et al., 2017)." has been added.

2) On lines 362-365, the authors state "Monitoring T_Opaque on longterm should provide important information which should help to better understand the LW cloud feedback mechanism. Moreover, because the relationship is linear, it simplifies the derivatives in mathematical expressions of feedback and will allow to construct a useful framework to study LW cloud feedback in simulations of climate models." Feedbacks are conventionally defined as the change in a given quantity holding all else fixed. In the case of altitude feedback, this would be the change in cloud altitude only, with everything including the temperature profile fixed. Mathematically, this is equivalent to comparing a control OLR with a hypothetical one computed with the cloud at a higher altitude and therefore at a lower emission temperature. Of course we know that in reality the cloud top temperature is expected to stay nearly constant with surface warming as the cloud top altitude rises with the isotherms (i.e., FAT hypothesis of Hartmann and Larson 2002). Changes in T_Opaque will depend on both the change in cloud altitude and the change in temperature profile, and constant T_Opaque may mean perfectly complementary changes in both the altitude and the temperature profile, as one expects from FAT. If one uses your relationship between OLR and T_Opaque in computing feedbacks, then the mathematical formulation of the feedbacks will need to be changed to accommodate this. Specifically, I think one would need to compare the fixed T_Opqaue (FAT) case against a hypothetical baseline situation in which all things change except for the Z_Opaque, such that T_Opaque warms as much as a fixed altitude. While this is do-able, I disagree with the statement above that this simplifies the mathematics of feedbacks.

- **Response:** We agree with the reviewer that it does not simplify the mathematics of feedbacks as the equation is currently as a function of $T_{Opaque}$. We will adapt this equation for a future study using climate model outputs with lidar simulator so that the equation will be as a function of the altitude of $T_{Opaque}$ ($Z_{T_{Opaque}}$) considering a linear atmospheric temperature lapse rate. In that way, a change in $Z_{T_{Opaque}}$, holding all else fixed, changes $CRE_{Opaque}$ by a quantity which, divided by the global mean raise in surface temperature, is directly the cloud altitude feedback. This will so simplify the mathematics of feedbacks.

- **Change made:**
  - In Sect. 4.2 (2nd §): "Moreover, because the relationship is linear, it simplifies the derivatives in mathematical expressions of feedback and will allow to construct a useful framework to study LW cloud feedback in simulations of climate models." has been removed.

3) The English is very poor throughout the manuscript. There were far too many errors for me to list all of them (grammar, spelling, awkward phrasings, words that are plural that should not be, incorrect comma usage, etc.). In some places the writing was poor enough that the meaning of the sentence was unclear. This paper should be copyedited by a native English speaker before the reviewers see it again. In contrast, the figures were very clear, well-designed, and well-executed.

- **Response:** A native English speaker copy-edited the paper.

**Minor Comments:** In addition to the numerous English errors, I note the following:

Title: I would suggest deleting "the" before Outgoing and also rephrasing to ". . .where a space borne-lidar. . ."

- **Change made:**

➤ **Title**: "Link between the Outgoing Longwave Radiation and the altitude where the space-borne lidar beam is fully attenuated" has been replaced by "The link between Outgoing Longwave Radiation and the altitude where a spaceborne lidar beam is fully attenuated ".

Throughout: "cloud altitude longwave" seems awkward. Please rephrase to "longwave cloud altitude"

- **Change made:**
  ➤ Throughout the paper: "cloud altitude longwave" has been replaced by "longwave cloud altitude".

Abstract: This ends very abruptly. It needs a better closing sentence.

- **Change made:**
  ➤ In Abstract: "The link between outgoing longwave radiation and the altitude where a spaceborne lidar beam is fully attenuated provides a simple formulation of the cloud radiative effect in the longwave domain and so helps to understand the longwave cloud altitude feedback mechanism." has been put as closing sentence.

Lines 29-34: An uninformed reader of this paragraph will assume that the only reason there is uncertainty in how clouds will respond to warming is because models simulate biased clouds in the mean state. Surely this is not the only reason for low confidence in cloud feedbacks. There are a variety of recent review articles out on cloud feedbacks that may be helpful on this point.

- **Response:** We agree with the reviewer. Thank you for this comment.
- **Change made:**
  ➤ In Introduction (1$^{st}$ §): "One reason for this uncertainty is that […]" has been added.

Lines 52-54: This statement needs to be rephrased. Emergent constraints are not feedback mechanisms.

- **Response:** We agree with the reviewer.
- **Change made:**
  ➤ In Introduction (3$^{rd}$ §): "Such records do not exist yet. Klein and Hall (2015) suggested that some cloud feedback mechanisms, namely the "emergent constraints", could be tested with shorter records in comparing the simulated and the observed current climate interannual variabilities" has been replaced by "Such records do not exist yet, but existing records might help our understanding (Klein and Hall, 2015).".

Lines 64-65: I disagree that there is no link between observed cloud variables and LW CRE. See, for example, the section on LW cloud altitude feedback in Ceppi et al doi: 10.1002/wcc.465 (2017), which points out that high cloud amount and emissivity, along with the temperature structure of the upper troposphere, govern the strength of this feedback. All of these are observable.

- **Response:** We wanted to focus on the fact that, so far, there was no simple mathematical expression to directly link, at different scales, cloud properties to OLR.
- **Change made:**
  ➤ In Introduction (4$^{th}$ §): "Nevertheless, the cloud altitude LW feedback mechanism and its amplitude still struggle to be verified in observations. There is still no observational confirmation for the altitude LW cloud feedback mechanism because 1) there is no simple direct and robust formulation linking the observed fundamental cloud variables and the LW CRE at the TOA […]" has been replaced by "Nevertheless, the LW cloud altitude feedback mechanism and its magnitude still remain to be confidently verified with observations, because 1) there is no simple, robust, and comprehensive mathematical formulation linking the observed fundamental cloud variables and the LW CRE at the TOA […]".

Lines 85-87: Cloud fraction histograms from passive sensors generally report cloud fraction on 7 cloud top pressure bins; the high, mid, and low aggregating is usually done later to simplify.

- **Response:** We agree with the reviewer.
- **Change made:**
  ➤ In Introduction (7$^{th}$ §): "[…] and only retrieve the cloud top pressure and estimates of high-level, mid-level, and low-level cloud covers. These last estimates have been coupled with ranges of cloud optical depth to define different cloud types (Hartmann et al., 1992) associated to different values of CRE." has been replaced by "[…] and instead retrieve single-layer effective cloud heights, often summarized as cloud fraction in seven cloud top pressure bins. Hartmann et al. (1992) used these pressure bins coupled with ranges of cloud optical depth to define different cloud types associated to different values of CRE.".

Lines 88-89: Suggest also citing Zhou et al DOI: 10.1175/JCLI-D-12-00547.1 (2013) and Yue et al 10.1002/2016JD025174 (2017), who have done this globally

- **Response:** Thank you for this suggestion.
- **Change made:**
  - In Introduction (7th §): "Zhou et al., 2013" and "Yue et al., 2017" have been added.

Lines 90-91: These studies should be more clearly distinguished from the ones preceding it in the sentence: they have focused on trends, not interannual variability.

- **Response:** We agree with the reviewer.
- **Change made:**
  - In Introduction (7th §): "[…], as well as the International Satellite Cloud Climatology Project (ISCCP) and the Pathfinder Atmospheres Extended (PATMOS-x) (Marvel et al., 2015; Norris et al., 2016) in order to identify LW CRE changes associated to cloud properties changes." has been replaced by "[…]. Recently, Marvel et al. (2015) and Norris et al. (2016) analyzed data from the International Satellite Cloud Climatology Project (ISCCP) and the Pathfinder Atmospheres Extended (PATMOS-x) datasets in terms of these cloud types to search for trends in LW CRE which would be associated with changes in cloud properties.".

Line 97: Mace et al (2011) DOI: 10.1175/2010JCLI3517.1 should be cited here

- **Response:** We agree with the reviewer.
- **Change made:**
  - In Introduction (8th §): "Mace et al., 2011" has been added.

Lines 168-170: I can't understand this. Please rephrase.

- **Change made:**
  - In Sect. 2.1 (3rd §): "Thin cloud emissivity $\varepsilon_{Thin}^{|}$ of a *Thin cloud single column* is inferred from the mean attenuated scattering ratio of levels flagged as "Clear" below the cloud, that we note $\langle SR' \rangle_{below}$ and which approximately corresponds to the apparent two-way transmittance through the cloud. Indeed, considering a fixed multiple scattering factor $\eta = 0.6$, we retrieve the Thin cloud visible optical depth $\delta_{Thin}^{VIS}$ (Garnier et al., 2015)." has been replaced by "Thin cloud emissivity $\varepsilon_{Thin}^{|}$ of a *Thin cloud single column* is inferred from the attenuated scattering ratio of clear sky layers measured by the lidar below the cloud. This is approximately equal to the apparent two-way transmittance through the cloud which, considering a fixed multiple scattering factor $\eta = 0.6$, allows retrieval of the Thin cloud visible optical depth $\tau_{Thin}^{VIS}$ (Garnier et al., 2015). As cloud particles are much larger than the wavelengths of visible and infrared light, and assuming there is no absorption by cloud particles in the visible domain, the Thin cloud LW optical depth $\tau_{Thin}^{LW}$ is approximately half of $\tau_{Thin}^{VIS}$ (Garnier et al., 2015).".

Line 183: should be "sea ice"

- **Change made:**
  - In Sect. 2.1 (last §): "iced sea" has been replaced by "sea ice".

Line 185: Should be "Flux observations collocated with lidar cloud observations"

- **Change made:**
  - In Sect. 2.2 (title): "Fluxes observations collocated with lidar clouds observations" has been replaced by "Flux observations collocated with lidar cloud observations".

Line 216: Should "as" be "that"?

- **Response:** Yes, indeed. Thank you.
- **Change made:**
  - In Sect. 3 (1st §): "such as" has been replaced by "such that".

Figure 4: Is it possible to compare these cloud emission temperatures with those from passive sensors? They should be in agreement, right?

- **Response:** Passive sensors do not allow a clear separation of Opaque clouds and Thin clouds as done with the lidar. Moreover, it does not find the same cloud occurrence. Cloud emissivity retrieval depends on hypothesis on clear sky and surface properties. Comparison with classical product derived from passive sensor is not obvious. An equivalent comparison was done by Stubenrauch et al. (2010) with collocated measurements from CALIOP and the passive sounder AIRS: they compared the height of the cloud emission temperatures determined by AIRS with the "apparent middle" of the cloud sounded by CALIOP, which is actually our definition of where the emission temperature of the cloud is. They show very good agreement.

Line 273: "T_opaque among opaque clouds" is redundant. This sort of statement occurs throughout the document.

- **Response:** We agree this precision makes the reading difficult.
- **Change made:**
  - Throughout the paper: "among Opaque clouds" and "among Thin clouds" have been removed from the main text but left into figure captions and the 1st § of Sect. 3.2 to avoid misunderstanding.

Line 282: meaning of "mid-effect" is unclear

- **Change made:**
  - In Sect. 3.2 (2nd §): "These Opaque clouds will have a mid-effect on the local OLR," has been replaced by "The local radiative effect of these Opaque clouds is weaker than the effect if they were in tropical ascending regions.".

Line 288: "pick" should be "peak"

- **Response:** Thank you.
- **Change made:**
  - In Sect. 3.2 (3rd §): "pick" has been replaced by "peak".

Line 303: rephrase

- **Change made:**
  - In Sect. 3.2 (last §): "[…] emissivities of Thin clouds are usually small, and clouds with small emissivities have less impact on the OLR. This, once again, goes in the sense that the role that play Thin clouds on the total CRE should be significantly smaller than that of Opaque clouds." has been replaced by "[…] emissivities of Thin clouds are usually small, so they have little impact on the OLR and hence their contribution to CRE should be significantly smaller than that of Opaque clouds.".

Lines 422-423: Rephrase.

- **Change made:**
  - In Sect. 5.2 (1st §): "Interestingly, an inversion of cover predominance and colder temperature between Opaque and Thin clouds occurs around 30° latitude. " has been replaced by "There are always more Opaque clouds than Thin clouds in the extratropics (beyond 30° latitude) and they are colder than the Thin clouds. It is the opposite in the tropical belt: there are always more Thin clouds than Opaque clouds, and those are slightly warmer.".

Figure 8: Is the shading 2-sigma? Max to min?

- **Change made:**
  - In figure caption of Fig. 8: "(max to min)" has been added.
- **Additional change:**
  - Fig. 8 has been redrawn because an error in our script was discovered. During computation of annual means of $T_{Opaque}$, $T_{Thin}$, and $\varepsilon_{Thin}$ on 2°x2° boxes (before averaging zonally), means were not weighted by monthly mean cover, on 2°x2° boxes, of opaque and thin clouds. It is now fixed. Changes are quite small and do not affect the conclusions.

Line 433: "under the tropics" – rephrase

- **Change made:**
  - In Sect. 5.2 (2nd §): "under the tropics" has been replaced by "in the tropics".

Line 453: I don't know what this statement means.

- ▪ **Response:** .
- ▪ **Change made:**
  - ➢ In Sect. 5.2 (last §): "Also, since the expression used for Thin clouds seems to give coherent results for $CRE_{Thin}^{\boxplus\,(LID)}$, it could also be used in a future work to quantify the role of a change in $C_{Thin}^{\boxplus}$, $T_{Thin}^{\boxplus}$, and $\varepsilon_{Thin}^{\boxplus}$ in the variations of $CRE_{Thin}^{\boxplus\,(LID)}$." has been replaced by "However, since the OLR expression above Thin clouds is almost as good as for the Opaque clouds, it could also be used in a future work to quantify the impact of changes in $C_{Thin}^{\boxplus}$, $T_{Thin}^{\boxplus}$, and $\varepsilon_{Thin}^{\boxplus}$ on the variations of $CRE_{Thin}^{\boxplus\,(LID)}$.".

Lines 488-493: The authors seem to be implying that omega is the only variable on which the cloud properties and CRE depend, and that therefore knowing how omega change will tell one how cloud properties and CRE will change. This is incorrect, as has been discussed many times over, most notably by Bony et al DOI 10.1007/s00382-003- 0369-6 (2004) where this type of analysis originally appeared. While omega changes may strongly determine regional changes in cloud properties, when averaged over the entire tropics, it is the thermodynamic sensitivity of cloud propertiesÂă within omega bins that emerges as the dominant driver of cloud changes.

- ▪ **Response:** We agree with the reviewer.
- ▪ **Change made:**
  - ➢ In Sect. 5.3 (last §): "Because cloud properties seem to be invariants for dynamical regimes, a change in the tropics of the large-scale circulation should provide a change in the CRE predictable and linked to the spatial distribution (both covers and altitudes) of Opaque clouds and Thin clouds sounded by CALIOP. For example, under global warming, climate models suggest a narrowing of the ascending branch of the Hadley cell (e.g. Su et al., 2014), which means less convective regions and more subsiding regions and which should result in a decrease of the CRE predictable knowing the changes of $\omega_{500}$ all over the tropics." has been replaced by "Because cloud properties seem to be invariants for dynamical regimes between 20 hPa·day$^{-1}$ and -100 hPa·day$^{-1}$, a change in the tropics of the large-scale circulation should lead to a predictable change in the CRE in regions that stay in this range of dynamical regimes, linked to the spatial distribution (both covers and altitudes) of Opaque clouds and Thin clouds sounded by CALIOP. For example, general circulation models suggest that a warmer climate will see a narrowing of the ascending branch of the Hadley cell (e.g. Su et al., 2014), which means less convective regions and more subsiding regions. This should result in a predictable decrease of the CRE, knowing the changes of $\omega_{500}$ for some part of the tropics.".

Section 6.1: It is unclear whether this is actually an error source. The authors raise the issue then immediately downplay it. Is it a source of error? Have you actually performed a sensitivity study to determine with these assumptions matter?

- ▪ **Response:** It is a source of error. The worst case for this is the multi-layer scenario when an optically thin cloud overlap an optically opaque cloud. This is now discussed in the new subsection 6.2. We could certainly have slightly more precise results using a centroid temperature for every case but it will add complexity to our expressions. However, the aim of our study is to find a simple expression of the CRE by determining its main cloud variable driver, not to reach the maximum of accuracy in CRE estimation.
- ▪ **Change made:**
  - ➢ In Sect. 6: Sect. 6.2 Multi-layer cloud and broken cloud situations has been added.
  - ➢ In Appendix:
    - o Fig. A4 has been added. It shows the decomposition of Fig. 6 in "single-layer cloud" and "multi-layer cloud" situations. The main text refers to Fig. A4 in Sect. 6.2.
    - o Fig. A5 has been added. It shows improvement on $OLR_{Opaque}$ when considering multi-layer cloud in the computation of $T_{Opaque}$. The main text refers to Fig. A5 in Sect. 6.2.

Section 6.4: the impacts of these assumptions are being assessed on the global mean OLR, but I wonder whether they also influence the slope of OLR on T_Opaque.
- ▪ **Response:** As $Z_{Opaque}$ is increased in every profile with the same amount (+480 m), and because atmospheric temperature profile is linearly dependent on the altitude, the slope of $OLR_{Opaque}$ on $T_{Opaque}$ is not influenced.

**Point-by-point response to the reviews**
* * *
**Anonymous Referee #3**

General

The authors present a methodology to estimate the outgoing longwave radiation (OLR) at the global scale from cloud products derived with the help of long-term space-borne measurements with the lidar CALIOP onboard CALIPSO. The major information comes from the opacity altitude of the atmosphere, i.e. the altitude at which the laser beam is fully attenuated due to clouds, and the geometrical cloud top height, which together allow the estimation of the radiative temperature of the cloud. It is shown that the latter one is linearly related to the OLR. Non-opaque (thin) clouds are treated in terms of top and base heights together with their emissivity, which is estimated from the lidar attenuated scattering ratio below the cloud under consideration of a constant multiple-scattering factor. For opaque clouds, a very good correlation between the derived OLR and the one measured by CERES is found, whereas a systematic deviation is seen for thin clouds. Despite some possible explanations the reason for the deviation in the case of thin clouds does not finally become clear.

In general, the paper presents an interesting approach to study longwave radiation effects of clouds at the global scale. The paper deserves publication, but has the potential to be improved both in terms of scientific contents as well as style of presentation. I recommend publication after consideration of the comments below.

**Major**

My major concerns are related to the rather simplified approach of using only two cloud scenarios, namely single-layer thin and opaque clouds. I would at least expect an extended sensitivity study regarding more realistic scenes in the very beginning. Justifying the approach before the presentation and discussion of results would be much more satisfying for the reader than the currently provided discussion of limitations in Sec. 6 (where several questions are tackled which the reader has already in mind when reading the major part of the paper). In particular, the following cases need to be considered in the evaluation and discussion of obtained results throughout the paper, starting already in Sec. 2.1 and Fig. 1.

- ▪ **Response:**
  - – We agree with the reviewer that the proposed cases need to be discussed. We have dedicated a subsection in Sect. 6 for this. Specifications are given below.
  - – We first tried to discuss all these aspects throughout the paper. However, this approach drowns the main message of the paper in digression. This is why we have decided to summarize them in Sect. 6.
- ▪ **Change made:**
  - ➢ In Sect. 6: Sect. 6.2 Multi-layer cloud and broken cloud situations has been added.

1) Multi-layer clouds: The discussion related to multi-layer clouds is not sufficient. The authors have added a very short paragraph in Sec. 2.1 (lines 176-179) during the technical revision of the paper. However, this explanation deals with thin clouds only. The more common feature is the appearance of thin, high cirrus clouds over mid-level or low-level opaque clouds. It is well known that retrievals from passive sensors locate the radiative cloud top height (or radiative temperature) in between the cloud layers in such cases, and that the location will depend on the optical thickness of the upper "thin" cloud. This fact is obviously not covered by the presented approach, since it considers only the geometrical properties of cloud top height and opacity altitude for the calculation of the radiative temperature. Although some discussion is provided in Sec. 6.2, no substantial investigation of the related consequences for the approach is given.

- ▪ **Response:** Thank you for this comment. An extensive investigation of multi-layer clouds is now given in Sect. 6.2.
- ▪ **Change made:**
  - ➢ In Appendix:
    - o Fig. A4 has been added. It shows the decomposition of Fig. 6 in "single-layer cloud" and "multi-layer cloud" situations. The main text refers to Fig. A4 in Sect. 6.2 (1st §).
    - o Fig. A5 has been added. It shows improvement on $OLR_{Opaque}$ when considering multi-layer cloud in the computation of $T_{Opaque}$. The main text refers to Fig. A5 in Sect. 6.2 (1st §).

2) Broken clouds: The authors find a high amount of "thin clouds" in the lower troposphere at temperatures above 0 ∘C, i.e. liquid clouds (Fig. 4). Usually, liquid clouds are not penetrated by lidar, even if they are geometrically thin (thickness of a few hundred meters). Those occasions of "thin clouds" might often be related to broken opaque clouds partly hit and partly missed by the lidar beam, thus leading to signals from the cloud and from the atmosphere and surface below the cloud in the same profile, so that the cloud appears to be transparent. The effect may be due to broken clouds within a single laser footprint, but can also result from averaging of laser shots over cloudy and clear atmospheric volumes before further retrievals are applied. From the description in Sec. 2.1, it does not become clear how averaging of lidar profiles is done, what exactly is meant with "each atmospheric single column" (line 127), which basic products (single shot, 1-km averages, 5-km averages) are used, and how the averaging to the 2◦x2◦ grid is performed. It should be studied which differences in the results are expected when sub-scale broken opaque clouds instead of thin clouds appear. It would be interesting to see whether the worse correlation between calculated and measured OLR found for thin clouds could be explained in this way. In this context, also the discussion in Sec. 6.2 is insufficient.

- ▪ **Response:** We agree with the reviewer that broken opaque clouds can be classified as thin clouds. However, in the GOCCP product, we do not average lidar profiles horizontally, so we only use single shot (90 m diameter footprint), which minimize this misclassification. Moreover, we plotted same as Fig. 6b only for Thin clouds with $T_{Thin} > 0\ °C$ (see Fig. A6): it shows excellent agreement between observed and lidar-derived OLR, and so does not explain the worse correlation between calculated and measured OLR as these clouds show excellent agreement.
- ▪ **Change made:**
  - ➤ In Sect. 2.1 (1$^{st}$ §): "The GCM-Oriented CALIPSO Cloud Product (GOCCP)-OPAQ (GOCCP v3.0; Guzman et al., 2017) segregates each atmospheric single column sounded by the CALIOP lidar as one of the 3 following single column types" has been replaced by "The GCM-Oriented CALIPSO Cloud Product (GOCCP)-OPAQ (GOCCP v3.0; Guzman et al., 2017) has 40 vertical levels with 480 m vertical resolution. Every CALIOP single shot profile — including multi-layer profiles — is classified into one of three types".
  - ➤ In Appendix:
    - ○ Fig. A6 has been added. It shows Fig. 6b but only with $T_{Thin}^{\varnothing} > 0\ °C$. The main text refers to Fig. A6 in Sect. 6.2 (2$^{nd}$ §).

**Minor**

Abstract: The abstract doesn't say anything about the retrievals for thin clouds.

- ▪ **Change made:**
  - ➤ In Abstract: "Similarly, the longwave cloud radiative effect of optically thin clouds can be derived from their top and base altitudes and an estimate of their emissivity." has been added.

Line 185, should be: "Flux observations collocated with lidar cloud observations"

- ▪ **Change made:**
  - ➤ In Sect. 2.2 (title): "Fluxes observations collocated with lidar clouds observations" has been replaced by "Flux observations collocated with lidar cloud observations".

Line 290, regarding the "second mode": What does "more diffuse" mean? What about altocumulus, altostratus clouds?

- ▪ **Response:** We agree the reviewer it could also be due to altocumulus or altostratus clouds.
- ▪ **Change made:**
  - ➤ In Sect. 3.2 (3$^{rd}$ §): "The second mode could be due to more diffuse or developing convective clouds." has been replaced by "The middle mode, near 5 km, might be due to developing convective clouds or middle altitude clouds.".

Line 300, "cloud emissivity of the cloud": correct to either "cloud emissivity" or "emissivity of the cloud".

- ▪ **Response:** Thank you.
- ▪ **Change made:**
  - ➤ In Sect. 3.2 (last §): "cloud emissivity of the cloud" has been replaced by "cloud emissivity".

Lines 331-332, "in spite of significant differences in the atmospheric temperature and humidity profiles": What does "significant" mean? How are these differences considered/validated in the calculations?

- ▪ **Response:** The sentence was indeed not very clear, it has been modified.
- ▪ **Change made:**

> ➢ In Sect. 4.1 (2nd §): "Linear regressions done on other regions with different atmospheric conditions give a similar coefficient. This means that, in spite of the significant differences in the atmospheric temperature and humidity profiles, $OLR^\downarrow_{Opaque}$ depends essentially only on $T^\downarrow_{Opaque}$." has been replaced by "Conducting the same linear regression on very different atmospheric conditions (from tropical to polar) gives similar coefficients. This means that $OLR^\downarrow_{Opaque}$ depends mainly on $T^\downarrow_{Opaque}$.".

Line 372, "The evaluation . . . is only using observation from January 2008": This explanation should be given in the beginning of the discussion of Fig. 6.

- ▪ **Response:** .
- ▪ **Change made:**
  - ➢ In Sect. 4.2 (2nd §): "Figure 6 compares lidar-derived and observed OLR during January 2008." has been added.
  - ➢ In Sect. 4.2 (last §): "The evaluation showed in Fig. 6 is only using observation from January 2008." has been removed.

Lines 405-415: Explain the units to be applied in the equations.

- ▪ **Change made:**
  - ➢ In Sect. 5.1: "[…] where $CRE^{\boxplus\,(LID)}_{Opaque}$ and $OLR^{\boxplus}_{Clear}$ are expressed in W·m-2 and $T^{\boxplus}_{Opaque}$ in K." and "[…] where $CRE^{\boxplus\,(LID)}_{Thin}$ and $OLR^{\boxplus}_{Clear}$ are expressed in W·m-2 and $T^{\boxplus}_{Thin}$ in K." have been added after Eqs. (7) and (8).

Lines 556 and 561, "decreases...from...", "reduces...from...": The meaning of the sentences with the word "from" is unclear.

- ▪ **Change made:**
  - ➢ In Sect. 6.4: "from" has been replaced by "by".

There a many language/grammar/punctuation errors, which cannot be listed in detail here. The manuscript needs careful copy editing.

- ▪ **Response:** A native English speaker copy-edited the paper.

[revised manuscript text omitted]
 ofDefining a simple and robust linear formulation between linking the LW CRE at the TOA and to a limited number of cloud variables, that would be more directly useful for decomposing climate cloud climate feedbacks decomposition. This formulation, however, cannot use utilize the detailsed of the entire cloud vertical distribution: first, one needs to but must be summarize the entire cloud vertical profile within a fewbased on specific cloud levels that drives the

LW CRE at the TOA, and second, this. Further, these specific cloud levels need tomust be accurately observed observable at global scale from satellites.

Most of the cloud climatologies derived from space observations rely on passive satellites, which do not retrieve the actual detailed cloud vertical distribution, and only retrieve the cloud top pressure and estimates of high-level, mid-level, and low-level cloud coversinstead retrieve single-layer effective cloud heights, often summarized as cloud fraction in seven cloud top pressure bins. These last estimates have been Hartmann et al. (1992) used these pressure bins coupled with ranges of cloud optical depth to define different cloud types (Hartmann et al., 1992)associated to different values of CRE. These cloud types have been used to analyze the interannual cloud record collected by the Moderate Resolution Imaging

Spectroradiometer (MODIS) (e.g. Zelinka and Hartmann, 2011; Zhou et al., 2013; Yue et al., 2017)., Recently, Marvel et al.

(2015) and Norris et al. (2016) analyzed data from as well as the International Satellite Cloud Climatology Project (ISCCP)

and the Pathfinder Atmospheres Extended (PATMOS-x) datasets (Marvel et al., 2015; Norris et al., 2016)in terms of these cloud types order to identifysearch for trends in LW CRE changeswhich would be associated to with changes in cloud properties changes.

[revised manuscript text omitted]

(a) $OLR_{Total}^{\boxplus(LID)}$ $(242.7\ W\cdot m^{-2})$

(b) $OLR_{Total}^{\boxplus(CERES)}$ $(242.8\ W\cdot m^{-2})$

(c) $OLR_{Total}^{\boxplus(LID)} - OLR_{Total}^{\boxplus(CERES)}$ $(-0.1\ W\cdot m^{-2})$

FIG. 7. Comparison between observed and lidar-derived OLR at 2°×2° gridded scale: (a) derived from CALIOP observations and (b) measured by CERES-Aqua. (c) = (a) - (b). Only  nighttime conditions over ice-free oceans for the 2008–2010 period  are considered. Global mean values are given in parentheses.

**5 Contributions of Opaque clouds and Thin clouds to the cloud radiative effect**

In the previous section, we found a  linear relationship  between $OLR_{Opaque}$ and $T_{Opaque}$ at different scales. The relationship for Thin clouds, though quite simple, is not linear and agrees less with observations than for Opaque clouds. In this section, we evaluate the contributions of Opaque clouds and Thin clouds to the total CRE.

**5.1 Partitioning cloud radiative effect into Opaque CRE and Thin CRE**

Using Eq. (5), we can decompose the total CRE at the TOA, computed from lidar observations, in  Opaque and Thin clouds contributions:

$$CRE_{Total}^{\boxplus\,(LID)} = OLR_{Clear}^{\boxplus} - OLR_{Total}^{\boxplus\,(LID)}$$

$$= C_{Opaque}^{\boxplus}\left(OLR_{Clear}^{\boxplus} - OLR_{Opaque}^{\boxplus\,(LID)}\right) + C_{Thin}^{\boxplus}\left(OLR_{Clear}^{\boxplus} - OLR_{Thin}^{\boxplus\,(LID)}\right). \tag{6}$$

$$\underbrace{\phantom{C_{Opaque}^{\boxplus}\left(OLR_{Clear}^{\boxplus} - OLR_{Opaque}^{\boxplus\,(LID)}\right)}}_{CRE_{Opaque}^{\boxplus\,(LID)}} \qquad \underbrace{\phantom{C_{Thin}^{\boxplus}\left(OLR_{Clear}^{\boxplus} - OLR_{Thin}^{\boxplus\,(LID)}\right)}}_{CRE_{Thin}^{\boxplus\,(LID)}}$$

Thereby, using Eq. (3), we can express $CRE_{Opaque}^{\boxplus\,(LID)}$ as a function of $C_{Opaque}^{\boxplus}$, $T_{Opaque}^{\boxplus}$ and $OLR_{Clear}^{\boxplus}$:

$$CRE_{Opaque}^{\boxplus\,(LID)} = C_{Opaque}^{\boxplus}\left(OLR_{Clear}^{\boxplus} - 2.0\,T_{Opaque}^{\boxplus} + 310\right)\,\cancel{.} \tag{7}$$

where $CRE_{Opaque}^{\boxplus\,(LID)}$ and $OLR_{Clear}^{\boxplus}$ are expressed in W·m$^{-2}$ and $T_{Opaque}^{\boxplus}$ in K.

Using Eq. (4), we can express $CRE_{Thin}^{\boxplus\,(LID)}$ as a function of $C_{Thin}^{\boxplus}$, $T_{Thin}^{\boxplus}$, $\varepsilon_{Thin}^{\boxplus}$ and $OLR_{Clear}^{\boxplus}$:

$$CRE_{Thin}^{\boxplus\,(LID)} = C_{Thin}^{\boxplus}\,\varepsilon_{Thin}^{\boxplus}\left(OLR_{Clear}^{\boxplus} - 2.0\,T_{Thin}^{\boxplus} + 310\right)\,\cancel{.} \tag{8}$$

where $CRE_{Thin}^{\boxplus\,(LID)}$ and $OLR_{Clear}^{\boxplus}$ are expressed in W·m$^{-2}$ and $T_{Thin}^{\boxplus}$ in K.

**5.2 Global means of the Opaque cloud CRE and the Thin cloud CRE**

Figure 8 shows the zonal mean observations of the 5 cloud properties ($C_{Opaque}^{\boxplus}$, $T_{Opaque}^{\boxplus}$, $C_{Thin}^{\boxplus}$, $T_{Thin}^{\boxplus}$ and $\varepsilon_{Thin}^{\boxplus}$).  Over the subsidence branches of the Hadley cell, around 20° S and 20° N, $C_{Opaque}^{\boxplus}$ is minimum (Fig. 8a), $T_{Opaque}^{\boxplus}$ and $T_{Thin}^{\boxplus}$ are warm (Fig 8b, temperatures in y-axis oriented downward) and $\varepsilon_{Thin}^{\boxplus}$ is minimum (Fig. 8c). So, we do not expect a  large contribution to the CRE from these regions. In contrast, the Intertropical Convergence Zone (ITCZ) corresponds to local maxima of Opaque and Thin cloud covers, extremely cold $T_{Opaque}^{\boxplus}$ and $T_{Thin}^{\boxplus}$ and a maximum of $\varepsilon_{Thin}^{\boxplus}$.  A large CRE is, therefore, expected from this region.  There are always more Opaque clouds than Thin clouds in the extratropics (beyond 30° latitude) and they are colder than the Thin clouds. It is the opposite in the tropical belt: there are always more Thin clouds than Opaque clouds, and those are slightly warmer. This suggests that the relative contribution of the Thin clouds to the CRE is larger in the tropic than in the rest of the globe. This should not be very dependent on  a specific year since the interannual variations of these 5 cloud properties (represented by the shaded areas) are very small compared to the zonal differences.

[Figure]

FIG. 8. Zonal mean observations: (a) $C_{Opaque}^{\boxplus}$ and $C_{Thin}^{\boxplus}$, (b) $T_{Opaque}^{\boxplus}$ among Opaque clouds and $T_{Thin}^{\boxplus}$ among Thin clouds and (c) $\varepsilon_{Thin}^{\boxplus}$ among Thin clouds. Only nighttime conditions over ice-free oceans for the 2008–2015 period is are considered. Shaded areas represent the envelope (max to min) including interannual variations.

Figure 9 shows that Opaque clouds contribute the most (73 %) to the total CRE. We can also note that the zonal variations of $CRE_{Opaque}^{\boxplus\,(LID)}$, and so approximately the variations of $CRE_{Total}^{\boxplus\,(LID)}$ (black line), can be explained by the zonal variations of $T_{Opaque}^{\boxplus}$ and $C_{Opaque}^{\boxplus}$ (Fig. 8a,b). For example, the absolute maximum $CRE$ at 5° N (~44 W·m$^{-2}$) is associated with a large cover and cold temperature of Opaque clouds. As suggested hereinbeforeearlier, we see that the relative contribution of Thin clouds ($CRE_{Thin}^{\boxplus\,(LID)}/CRE_{Total}^{\boxplus\,(LID)}$, Fig. 9b) is larger under in the tropics, approximately 2 timestwice larger below 30° (up to 40 %) than beyond those latitudes.

[Figure]

FIG. 9. (a) Partitioning of total CRE into Opaque CRE and Thin CRE. (b) Ratios of the Opaque and (Thin) CRE to the total CRE. Only nighttime conditions over ice-free oceans for the 2008–2015 period is are considered.

Figure 10 shows the same CRE partitioning on maps. The likeness similarity of patterns between total CRE (Fig. 10a) and the Opaque clouds CRE contribution (Fig. 10b) is prominentobvious, strengthening showing again the importance of thethat Opaque clouds in mostly drive the CRE. We can also note thatThe contribution of Thin clouds to the CRE contribution (Fig. 10c) have is 
[revised manuscript text omitted]
}}\left(\delta\tau^{LW|} = 0\right) = \int_0^{\delta\tau^{LW|}_{Cloud}} B_v\left(T\left(\delta\tau^{LW|}\right)\right)e^{-\delta\tau^{LW|}}d\delta\tau^{LW|} \qquad\qquad [\text{W·m}^{-2}\text{·sr}^{-1}\text{·m}^{-1}] \text{ (A1)}$$

Considering a linear increase of the temperature with $\delta\tau^{LW|}$ from the cloud top to the cloud base ($T\left(\delta\tau^{LW|}\right) = k_1\delta\tau^{LW|} + k_2$) and integrating $I^{|}_{v_{Cloud}}$ throughout the whole LW spectrum (using Stefan-Boltzmann law $\int B_v\,dv = \sigma T^4/\pi$), we can write the LW radiance $I^{LW|}$ emitted by the cloud at the top of the cloud as:

$$I^{LW|}_{Cloud}(\delta\tau^{LW|} = 0) = \int_0^{\delta\tau^{LW|}_{Cloud}} \frac{\sigma}{\pi}\left(k_1\delta\tau^{LW|} + k_2\right)^4 e^{-\delta\tau^{LW|}}d\delta\tau^{LW|} \qquad\qquad [\text{W·m}^{-2}\text{·sr}^{-1}]$$

(A2)

Assuming that the cloud emits as a Lambertian surface, the upward LW radiative flux $F^{\uparrow LW|}$ emitted by the cloud at the top of the cloud is given by:

$$F^{\uparrow LW|}_{Cloud}\left(\delta\tau^{LW|} = 0\right) = \int_0^{\delta\tau^{LW|}_{Cloud}} \sigma\left(k_1\delta\tau^{LW|} + k_2\right)^4 e^{-\delta\tau^{LW|}}d\delta\tau^{LW|} \qquad\qquad [\text{
[revised manuscript text omitted]